# Simultaneous Dimensionality Reduction: A Data Efficient Approach for Multimodal Representations Learning

**Eslam Abdelaleem**\*  *eslam.abdelaleem@emory.edu*
*Department of Physics*
*Emory University*

**Ahmed Roman**†  *ahmed.hemdan.roman@emory.edu*
*Department of Physics*
*Emory University*

**K. Michael Martini**  *karl.michael.martini@emory.edu*
*Department of Physics & Initiative in Theory and Modeling of Living Systems*
*Emory University*

**Ilya Nemenman**  *ilya.nemenman@emory.edu*
*Department of Physics & Department of Biology & Initiative in Theory and Modeling of Living Systems*
*Emory University*

**Reviewed on OpenReview:** *https://openreview.net/forum?id=Ni14fXbyTV*

## Abstract

Current experiments frequently produce high-dimensional, multimodal datasets—such as those combining neural activity and animal behavior or gene expression and phenotypic profiling—with the goal of extracting useful correlations between the modalities. Often, the first step in analyzing such datasets is dimensionality reduction. We explore two primary classes of approaches to dimensionality reduction (DR): Independent Dimensionality Reduction (IDR) and Simultaneous Dimensionality Reduction (SDR). In IDR methods, of which Principal Components Analysis is a paradigmatic example, each modality is compressed independently, striving to retain as much variation within each modality as possible. In contrast, in SDR, one simultaneously compresses the modalities to maximize the covariation between the reduced descriptions while paying less attention to how much individual variation is preserved. Paradigmatic examples include Partial Least Squares and Canonical Correlations Analysis. Even though these DR methods are a staple of statistics, their relative accuracy and data set size requirements are poorly understood. We use a generative linear model to synthesize multimodal data with known variance and covariance structures to examine these questions. We assess the accuracy of the reconstruction of the covariance structures as a function of the number of samples, signal-to-noise ratio, and the number of varying and covarying signals in the data. Using numerical experiments, we demonstrate that linear SDR methods consistently outperform linear IDR methods and yield higher-quality, more succinct reduced-dimensional representations with smaller datasets. Remarkably, regularized CCA can identify low-dimensional weak covarying structures even when the number of samples is much smaller than the dimensionality of the data, which is a regime challenging for all dimensionality reduction methods. Our work corroborates and explains previous observations in the literature that SDR can be more effective in detecting covariation patterns in data. These findings strengthen the intuition that SDR should be preferred to IDR in real-world data analysis when detecting covariation is more important than preserving variation.

---

\*Corresponding author
†Currently `roman@broadinstitute.org` at Department of Medical Oncology - Dana-Farber Cancer Institute & Broad Institute of MIT and Harvard & Harvard Medical School

## 1 Introduction

Many modern experiments across various fields generate massive multimodal data sets. For instance, in neuroscience, it is common to record the activity of a large number of neurons while simultaneously recording the resulting animal behavior (Stringer et al., 2019; Steinmetz et al., 2021; Urai et al., 2022; Krakauer et al., 2017). Other examples include measuring gene expressions of thousands of cells and their corresponding phenotypic profiles, or integrating gene expression data from different experimental platforms, such as RNA-Seq and microarray data (Clark et al., 2013; Zheng et al., 2017; Svensson et al., 2018; Huntley et al., 2015; Lorenzi et al., 2018). In economics, important variables such as inflation are often measured using combinations of macroeconomic indicators as well as indicators belonging to different economic sectors (Gosselin & Tkacz, 2001; Baillie et al., 2002; Freyaldenhoven, 2022; Rudd, 2020). In all of these examples, an important goal is to estimate statistical correlations among the different modalities.

Analyses usually begin with dimensionality reduction (DR) into a smaller and more interpretable representation of the data. We distinguish two types of DR: *independent* (IDR) and *simultaneous* (SDR) (Martini & Nemenman, 2024). In the former, each modality is reduced independently, while aiming to preserve its variation, which we call *self* signal. In the latter, the modalities are compressed simultaneously, while maximizing the covariation (or the *shared* signal) between the reduced descriptions and paying less attention to preserving the individual variation. It is not clear if IDR techniques, such as the Principal Components Analysis (PCA) (Hotelling, 1933), are well-suited for extracting shared signals since they may overlook features of the data that happen to be of low variance, but of high covariance (Colwell et al., 2014; Borga et al., 1997). In particular, poorly sampled weak shared signals, common in high-dimensional datasets, can exacerbate this issue. SDR techniques, such as Partial Least Squares (PLS) (Wold et al., 2001) and Canonical Correlations Analysis (CCA) (Hotelling, 1936), are sometimes mentioned as more accurate in detecting weak shared signal (Chin & Newsted, 1999; Hair et al., 2011; Pacharawongsakda & Theeramunkong, 2016). However, the relative accuracy and data set size requirements for detecting the shared signals in the presence of self signals and noise remain poorly understood for both classes of methods.

In this study, we aim to assess the strengths and limitations of linear IDR, represented by PCA, and linear SDR, exemplified by PLS and CCA, in detecting weak shared signals. For this, we use a generative linear model that captures key features of relevant examples, including noise, the self signal, and the shared signal components. Using this model, we analyze the performance of the methods in different conditions. We assess how well these techniques can (i) extract the relevant shared signal and (ii) identify the dimensionality of the shared and the self signals from noisy, undersampled data. We investigate how the signal-to-noise ratios, the dimensionality of the reduced variables, and the method of computing correlations combine with the sample size to determine the quality of the DR. We propose best practices for achieving high-quality reduced representations with small sample sizes using these linear methods.

Overall, the main contributions of this paper are:

- We define a tractable, generative, linear model for producing multimodal datasets; the model allows us to tune the numbers and the strengths of the shared and the self signals in the generated modalities.

- We characterize the accuracy and data set requirements for reconstructing the shared signals by SDR methods (CCA, rCCA, and PLS) and IDR methods (PCA) as a function of the parameters of the generative model.

- We find that SDR methods generally outperform IDR methods in detecting shared signals in multimodal data; this is true for both the synthetic generative linear model, as well for the non-linear data derived from the MNIST dataset.

## 2   Model

### 2.1   Relations to Previous Work

The extraction of signals from large-dimensional data sets is a challenging task when the number of observations is comparable to or smaller than the dimensionality of the data. The undersampling problem introduces spurious correlations that may appear as signals, but are, in fact, just statistical fluctuations. This poses a challenge for DR techniques, as they may retain unnecessary dimensions or identify noise dimensions as true signals. Here, we focus exclusively on linear DR methods. For these, the Marchenko-Pastur (MP) distribution of eigenvalues of the covariance matrix of pure noise derived using the Random Matrix Theory (RMT) methods (Marchenko & Pastur, 1967) has been used to introduce a cutoff between noise and true signal in real datasets. However, recent work (Fleig & Nemenman, 2022a) has shown that, when observations are a linear combination of uncorrelated noise and latent low-dimensional self signals, then the self signals alter the distribution of eigenvalues of the sampling noise, questioning the validity of this naive approach.

Moving beyond a single modality, Bouchaud et al. (2007) calculated the singular value spectrum of cross-correlations between two nominally uncorrelated random signals. However, it remains unknown whether the linear mixing of self signals and shared signals affects the spectra of noise, and how all of these components combine to limit the ability to detect shared signals between two modalities from data sets of realistic sizes. Filling in this gap using numerical simulations is the main goal of this paper, and analytical treatments of this problem are left for the future.

The linear model and linear DR approaches studied here do not capture the full complexity of real-world data sets and state-of-the-art algorithms. However, if sampling issues and self signals limit the ability of linear DR methods to extract shared signals, it would be surprising for nonlinear methods to succeed in similar scaling regimes on real data. Thus extending the previous work to explicitly study the effects of linear mixtures of self signals, shared signals, and noise on limitations of DR methods is likely to result in intuition transferable to more complex scenarios. Examples of such scenarios might include inference of dynamics of a system through a latent space (Creutzig et al., 2009; Chen et al., 2022), where shared signals correspond to latent factors that are relevant for predicting the future of the system from its past, while self signals correspond to nonpredictive variation (Bialek et al., 2001). In economics, shared and self signals correspond to diverse macroeconomic indicators that are grouped into correlated distinct categories in structural factor models (Forni & Gambetti, 2010; Gosselin & Tkacz, 2001; Rudd, 2020; Baillie et al., 2002). In neuroscience, shared signals can correspond to the latent space, by which neural activity affects behavior, while self signals encode neural activity that does not manifest in behavior and behavior that is not controlled by the part of the brain being recorded from (Sponberg et al., 2015; Stringer et al., 2019; Natraj et al., 2022; Sani et al., 2021; Pang et al., 2016; Urai et al., 2022; Krakauer et al., 2017).

Interestingly, in the context of the neural control of behavior, it was noticed that SDR reconstructs the shared neuro-behavioral latent space more efficiently and using a smaller number of samples than IDR (Sani et al., 2021). Similar observations have been made in more general statistical contexts (Chin & Newsted, 1999; Hair et al., 2011; Pacharawongsakda & Theeramunkong, 2016; Vogelstein et al., 2021), though the agreement is not uniform (Goodhue et al., 2006; 2012; 2013). Because of this, most practical recommendations for detecting shared signals are heuristic (Hair Jr et al., 2021), with widely acknowledged limitations (Kock & Hadaya, 2018). Our goal is to ground such rules in numerical simulations and scaling arguments.

### 2.2   Linear Model with Self and Shared Signals

We consider a linear generative model of data, which includes noise, $m_{\mathrm{self,X}}, m_{\mathrm{self,Y}}$ self signals that are relevant to each modality independently, as well as $m_{\mathrm{shared}}$ shared signals that capture the interrelationships

between modalities.[1] It results in $T$ observations of two standardized observables, $X$ and $Y$:

$$\begin{aligned}
\left[\tilde{X} \in \mathbb{R}^{T \times N_X}\right] &= \underbrace{R_X}_{\text{Independent white noise}} + \underbrace{U_X V_X}_{\text{Self-Signal for X}} + \underbrace{P Q_X}_{\text{Shared-Signal}}, \\
\left[\tilde{Y} \in \mathbb{R}^{T \times N_Y}\right] &= \underbrace{R_Y}_{\text{Independent white noise}} + \underbrace{U_Y V_Y}_{\text{Self-Signal for Y}} + \underbrace{P Q_Y}_{\text{Shared-Signal}}.
\end{aligned} \quad (1)$$

$$X = \tilde{X}/\sigma_{\tilde{X}}, Y = \tilde{Y}/\sigma_{\tilde{Y}}. \quad (2)$$

The observations of $X$ and $Y$ are linear combinations of the following: (a) independent white noise components $R_X \in \mathbb{R}^{T \times N_X}$ and $R_Y \in \mathbb{R}^{T \times N_Y}$ with variances $\sigma_{R_X}^2$ and $\sigma_{R_Y}^2$; (b) self-signal components $U_X$ and $U_Y$ residing in lower-dimensional spaces $\mathbb{R}^{T \times m_{\text{self},X}}$ and $\mathbb{R}^{T \times m_{\text{self},Y}}$, respectively, with the corresponding variances $\sigma_{U_X}^2$ and $\sigma_{U_Y}^2$; (c) shared-signal components $P$ in a shared lower-dimensional space $\mathbb{R}^{T \times m_{\text{shared}}}$, with variance $\sigma_P^2$. These lower-dimensional components are projected into their respective high-dimensional spaces $\mathbb{R}^{T \times N_X}$ and $\mathbb{R}^{T \times N_Y}$ using fixed quenched projection matrices $V_X \in \mathbb{R}^{m_{\text{self},X} \times N_X}$, $V_Y \in \mathbb{R}^{m_{\text{self},Y} \times N_Y}$, $Q_X \in \mathbb{R}^{m_{\text{shared}} \times N_X}$, and $Q_Y \in \mathbb{R}^{m_{\text{shared}} \times N_Y}$, with variances $\sigma_{V_X}^2$, $\sigma_{V_Y}^2$, $\sigma_{Q_X}^2$, and $\sigma_{Q_Y}^2$, respectively. Entries in these matrices are drawn from Gaussian distributions with zero means and the corresponding variances. Further, division by $\sigma_{\tilde{X}}$ and $\sigma_{\tilde{Y}}$ standardizes each column of the data matrices by their empirical standard deviations. The total variance in the matrix $\tilde{X}$ can be calculated as the sum of the variances of its individual components: $\sigma_{\tilde{X}}^2 = \sigma_{R_X}^2 + m_{\text{self},X} \times \sigma_{U_X}^2 \sigma_{V_X}^2 + m_{\text{shared}} \times \sigma_P^2 \sigma_{Q_X}^2$, and similarly for $\tilde{Y}$.

We define self and shared signal-to-noise ratios $\gamma_{\text{self},X/Y}, \gamma_{\text{shared},X/Y}$ as the relative strength of signals compared to background noise per component in each modality. These definitions allow us to examine how easily self or shared signals in each dimension can be distinguished from the noise.

$$\gamma_{\text{self},X/Y} = \frac{\sigma_{U_{X/Y}}^2 \sigma_{V_{X/Y}}^2}{\sigma_{R_{X/Y}}^2}, \quad \gamma_{\text{shared},X/Y} = \frac{\sigma_P^2 \sigma_{Q_{X/Y}}^2}{\sigma_{R_{X/Y}}^2}. \quad (3)$$

Our main goal is to evaluate the ability of linear SDR and IDR methods to reconstruct[2] the shared signal $P$, while overlooking the effects of the self signals $U_{X/Y}$ on the statistics of the shared ones. We formalize this goal in the next section by defining a shared signal reconstruction metric $\mathcal{RC}'$.

## 3 Methods

We apply DR techniques to $X$ and $Y$ to obtain their reduced dimensional forms $Z_X$ and $Z_Y$, respectively. $Z_X, Z_Y$ are of sizes that can range from $T \times 1$ to $T \times N_X$ and $T \times N_Y$, respectively. As an IDR method, we use PCA (Hotelling, 1933). As SDR methods, we apply PLS (Wold et al., 2001) and CCA (Hotelling, 1936; Vinod, 1976; Årup Nielsen et al., 1998), including both normal and regularized versions of the latter. Each of these methods focuses on specific parts of the overall covariance matrix

$$C_{X,Y} = \begin{bmatrix} C_{XX} & C_{XY} \\ C_{YX} & C_{YY} \end{bmatrix} = \begin{bmatrix} \frac{1}{T} X^\top X & \frac{1}{T} X^\top Y \\ \frac{1}{T} Y^\top X & \frac{1}{T} Y^\top Y \end{bmatrix}. \quad (4)$$

PCA aims to identify the most significant features that explain the majority of the *variance* in $C_{XX}$ and $C_{YY}$, independently. PLS, on the other hand, focuses on singular values and vectors that explain the *covariance* component $C_{XY}$. Along the same lines, CCA aims to find linear combinations of $X$ and $Y$ that are responsible for the *correlation* $(C_{XY}/\sqrt{C_{XX}C_{YY}})$ between $X$ and $Y$ (Borga et al., 1997). See Appendix A.1 for a detailed description of these methods.

---

[1]This model is an extension of the model introduced by Fleig & Nemenman (2022a), and its probabilistic form has been studied by Murphy (2022). In its turn, the latter is an extension of work by Klami et al. (2012), and Bach & Jordan (2005). However, within this model, we focus on the intensive limit, common in RMT (Potters & Bouchaud, 2020), where the number of observations scales as the number of observed variables. This scenario is common in many real-world applications. To our knowledge, a treatment of the extensive regime to assess different DR methods as a function of various parameters of the system does not exist.

[2]In this context, we use "reconstruction" and "detection" interchangeably. Detection means that we are able to capture the shared signal, which is equivalent to its reconstruction, as the signal is represented by the correlation.

For every numerical experiment, we generate training and test data sets $(X_{\text{train}}, Y_{\text{train}})$ and $(X_{\text{test}}, Y_{\text{test}})$ according to Eqs. (1-2)[3]. We apply PCA, PLS, CCA, and regularized CCA (rCCA) to the training to obtain the singular directions $W_{X_{\text{train}}}$ and $W_{Y_{\text{train}}}$ for each method (see Appendix A.1). We then obtain the projections of the test data on these singular directions

$$Z_X = X_{\text{test}} W_{X_{\text{train}}},$$
$$Z_Y = Y_{\text{test}} W_{Y_{\text{train}}}. \tag{5}$$

Finally, we evaluate the *reconstructed correlations* metric $\mathcal{RC}'$, which measures how well these singular directions recover the shared signals in the data, corrected by the expected positive bias due to the sampling noise, see Appendix A.2 for details. $\mathcal{RC}' = 0$ corresponds to no overlap between the true and the recovered shared directions, and $\mathcal{RC}' = 1$ corresponds to perfect recovery.

## 4    Results

### 4.1    Results for The Generative Linear Model

We perform numerical experiments to explore the undersampled regime, $T \lesssim N_X, N_Y$. We use $T = \{100, 300, 1000, 3000\}$ samples, $N_X = N_Y = 1000$. We explore the case of one shared signal only, $m_{\text{shared}} = 1$ and we mask this shared signal by a varying number of self signals and by noise. We vary the number of retained dimensions, $(|Z_X|, |Z_Y|$[4], and explore how many of them are needed to recover the shared signal in the noise and the self signal background with different SNRs.

For brevity, we explore two cases: (1) one self-signal each in $X$ and $Y$ in addition to the shared signal ($m_{\text{self}} = 1$); (2) many self-signals in $X$ and $Y$. For both cases, we calculate the quality of reconstruction as the function of the shared and the self SNR, $\gamma_{\text{shared}}$ and $\gamma_{\text{self}}$. In all figures, we show $\mathcal{RC}'$ for severely undersampled (first row, $T = 300$) and relatively well sampled (second row, $T = 3000$) regimes. We also show the value of $\mathcal{RC}_0$, the bias that we removed from our reconstruction quality metric, for completeness, see Appendix A.2 for details. Experiments at different parameter values can be found in Appendix A.4.

Figure 1 shows that, in Case 1, when one dimension is retained in DR of $X$ and $Y$, PCA populates the compressed variable with the largest variance signals and hence struggles to retain the shared signal when $\gamma_{\text{self}} > \gamma_{\text{shared}}$, regardless of the number of samples[5]. However, both PLS and rCCA excel in achieving nearly perfect reconstructions. When $T \ll N_X$, straightforward CCA cannot be applied (see A.1.3-A.1.4), but it too achieves a perfect reconstruction when $T > N_X$.

In Fig. 2, we allow two dimensions in the reduced variables. For PCA, we expect this to be sufficient to preserve both the self and the shared signals. Indeed, PCA now works for all $\gamma$s and $T$s, although with a slightly reduced accuracy for large shared signals compared to Fig. 1. PLS and rCCA continue to deliver highly accurate reconstructions. So does the CCA for $T > N_X$. Spurious correlations, as measured by $\mathcal{RC}_0$ grow slightly with the increasing dimensionality of $Z_X$, $Z_Y$ compared to Fig. 1. This is expected since more projections must now be inferred from the same amount of data.

We now turn to $m_{\text{self}} \gg m_{\text{shared}}$. We use $m_{\text{shared}} = 1$, $m_{\text{self}} = 30$ for concreteness. We expect that the performance of SDR methods will degrade weakly, as they are designed to be less sensitive to the masking effects of the self signals. In contrast, we expect IDR to be more easily confused by the many strong self-signals, as IDR is designed to pick up any large variance components, which would include self signals if they are stronger than the shared signal, degrading the performance. Indeed, Fig. 3 shows that PCA now faces challenges in detecting shared signals, even when the self signals are weaker than in Fig. 1. Increasing

---

[3]We fix $\sigma^2_{R_{X/Y}}, \sigma^2_{V_{X/Y}}, \sigma^2_{Q_{X/Y}}$ and allow $\sigma^2_{U_{X/Y}}, \sigma^2_P$ to vary –as equally spaced values between 0.05 and 1– when we choose $\gamma_{\text{self},X/Y}, \gamma_{\text{shared},X/Y}$. The SNRs of $X$ and $Y$ are symmetric. We first generate the fixed projection matrices $V_{X/Y}, Q_{X/Y}$, and we vary $R_{X/Y}, U_{X/Y}, P$ for each trial.

[4]We use the notation $|\cdot|$ to specify the dimensionality (or the number of dimensions we are keeping after reduction) of a variable. We reserve the usage of the notation $||\cdot||$ to the matrix norm, as used in defining the $\mathcal{RC}$ metric (c.f. A.2)

[5]The transition from being able to observe the shared signal to not being able to observe it in data as a function of the sample size resembles the famous BBP transition (Baik et al., 2004), which can be explored analytically for random matrices of the form Eqs. 1, which we leave for future work.

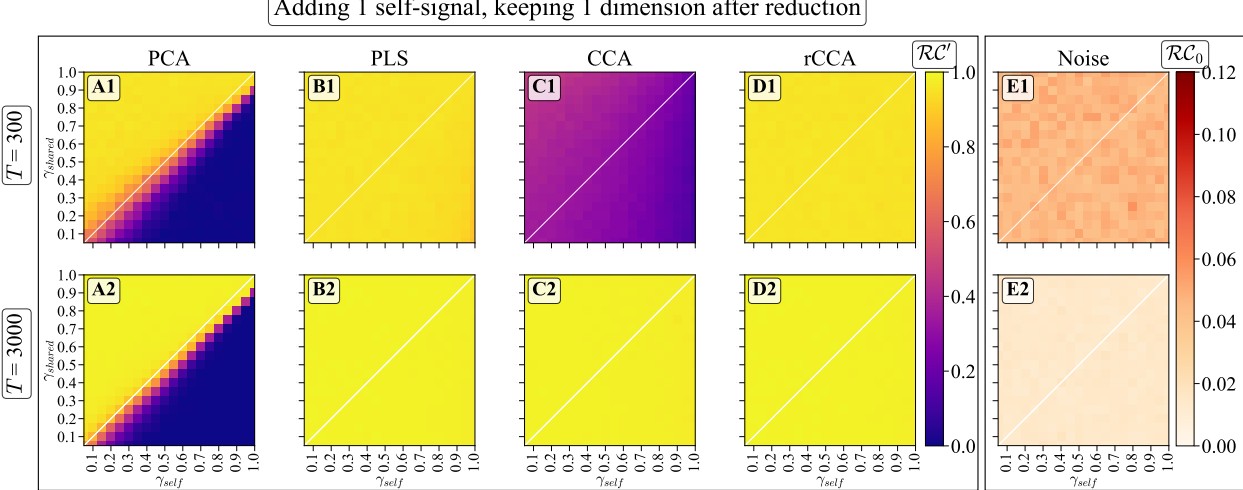

Figure 1: Performance of PCA, PLS, CCA, rCCA, and noise in recovery of the shared signal for $|Z_X| = |Z_Y| = 1 = m_{\text{self}}$. The rows are undersampled (top) and relatively well-sampled (bottom) scenarios, respectively. PCA struggles to detect shared signals when they are weaker than the self signals, even with more samples. PLS and rCCA demonstrate nearly perfect reconstruction. CCA displays no reconstruction in the undersampled regime $T \ll N_X$, and it is nearly perfect for large $T$.

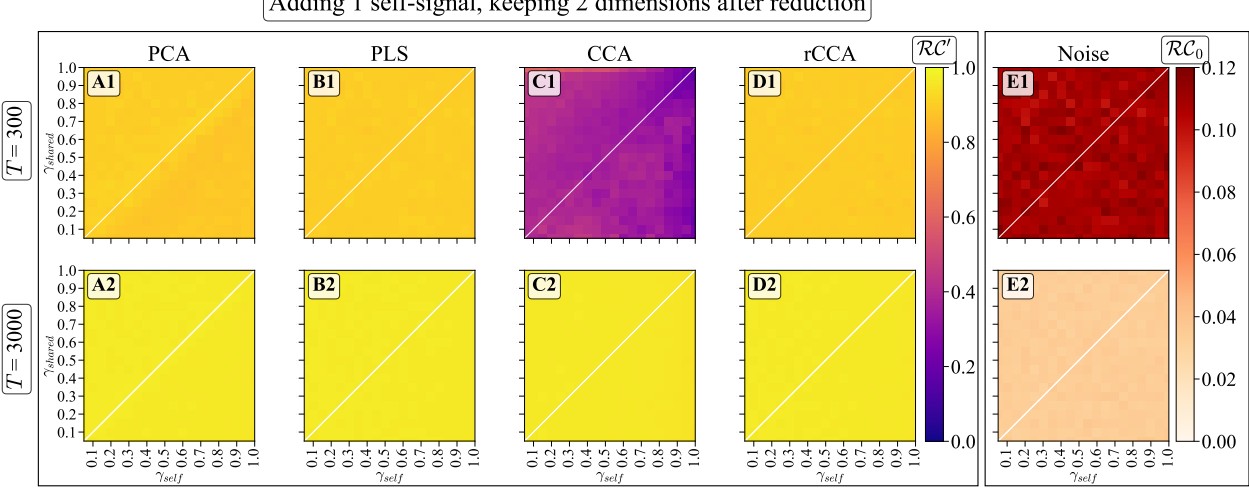

Figure 2: Same as Fig. 1, but for $|Z_X| = |Z_Y| = 2 = m_{\text{self}} + m_{\text{shared}}$. Now there are enough compressed variables for PCA to detect the shared signal. Other methods perform similarly to Fig. 1, albeit the noise is larger.

$T$ improves its performance only slightly. Somewhat surprisingly, PLS performance also degrades, with improvements at $T \gg N_X$. CCA again displays no reconstruction when $T \ll N_X$, switching to near perfect reconstruction at large $T$. Crucially, rCCA again shines, maintaining its strong performance, consistently demonstrating nearly perfect reconstruction.

Since one retained dimension is not sufficient for PCA to represent the shared signal when $\gamma_{\text{shared}} \lesssim \gamma_{\text{self}}$, we increase the dimensionality of reduced variables, $|Z_X| = |Z_Y| = m_{\text{self}} \gg m_{\text{shared}}$, cf. Fig. 4. PCA now detects shared signals even when they are weaker than the self-signals, $\gamma_{\text{shared}} < \gamma_{\text{self}}$, but at a cost of the reconstruction accuracy plateauing significantly below 1. In other words, when self and shared signals

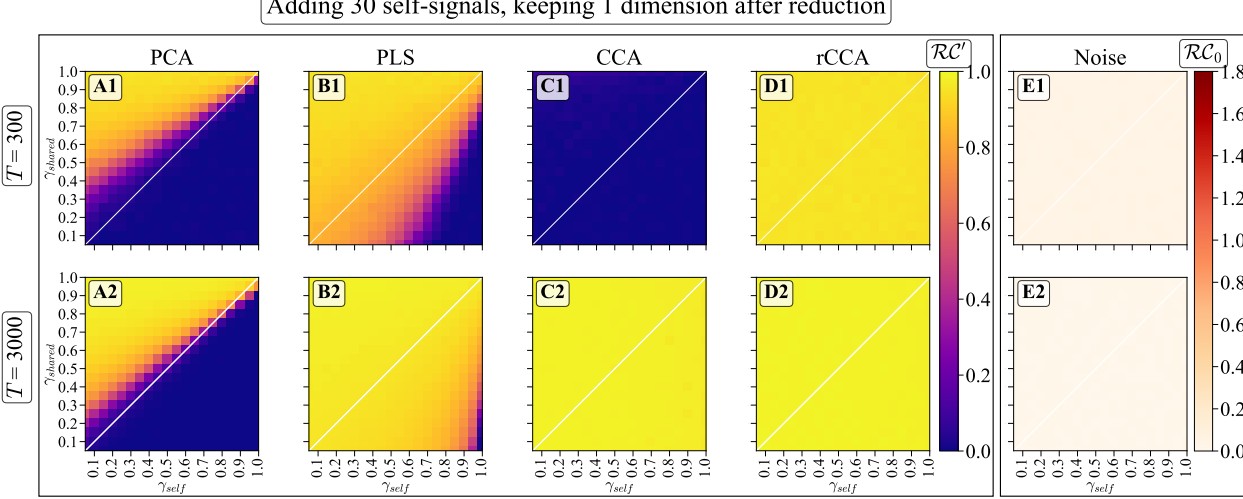

Figure 3: Reconstruction results for $m_{\text{self}} = 30$, $m_{\text{shared}} = 1$, and $|Z_X| = |Z_Y| = 1$. PCA struggles to detect any shared signals when they are even comparable to the self ones. PLS performance also degrades. CCA displays its usual impotence at small $T$. Finally, rCCA demonstrates nearly perfect reconstruction for all parameter values.

are comparable, they mix, allowing for partial reconstruction. However, even at $T \gg N_X$, PCA cannot break into the phase diagram's lower right corner. Other methods perform similarly, reconstructing shared signals over the same or wider ranges of sampling and the SNR ratios than in Fig. 3. For all of them, the improvement comes at the cost of the decreased asymptotic performance. The most distinct feature of this regime is the dramatic effect of noise, where 30-dimensional compressed variables can accumulate enough sampling fluctuations to recover correlations that are supposedly nearly twice as high as the data actually has.

Figure 5 now explores a regime when the dimensionality of the compressed variables is enough to store both the self and the shared interactions at the same time, $|Z_X| = |Z_Y| = m_{\text{self}} + m_{\text{shared}} = 31$. With just one more dimension than Fig. 4, PCA abruptly transitions to being able to recover shared signals for all SNRs, albeit still saturating at a far from perfect performance at large $T$. PLS, CCA, rCCA, and noise show behavior remain similar to Fig. 4.

Our analysis suggests that there are three relevant factors that determine the ability of DR to reconstruct shared signals. The first is the strength of the shared and the self signals compared to each other and to noise. For brevity, in the following analysis, we fix $\gamma_{\text{self}}$ and define the ratio $\tilde{\gamma} = \gamma_{\text{shared}}/\gamma_{\text{self}}$ to represent this effect. The second factor affecting the performance is the ratio between the number of shared and self signals, denoted by $\tilde{m} = m_{\text{shared}}/m_{\text{self}}$. The third factor is the number of samples per dimension of the reduced variable, denoted by $\tilde{q} = T/|Z|$.

In Fig. 6, we illustrate how these parameters influence the performance of DR, $\mathcal{RC}'$. Each subplot varies $\tilde{q}$, while holding $T$ constant and changing $|Z_X|$. We compare the results of PCA (representing IDR) and rCCA (representing SDR). Each curve is averaged over 10 trials, with error bars indicating 1 standard deviation around the mean, using algorithmic parameters as described in Appendix A.3.

We see that the relative strength of signals, as represented by $\tilde{\gamma}$, plays a significant role in determining, which method performs better. If the shared signals are larger (bottom) both approaches work. However, for weak shared signals (top), SDR is generally more effective. Further, the ratio between the number of shared and self signals, $\tilde{m}$, also plays an important role. When $\tilde{m}$ is large (left), IDR is more likely to detect the shared signal before the self signals, and it approaches the performance of SDR. However, when $\tilde{m}$ is small, IDR is more likely to capture the self signals before moving on to the shared signals, degrading performance (right). Finally, not surprisingly, the number of samples per dimension of the compressed variables, $\tilde{q}$, is also critical

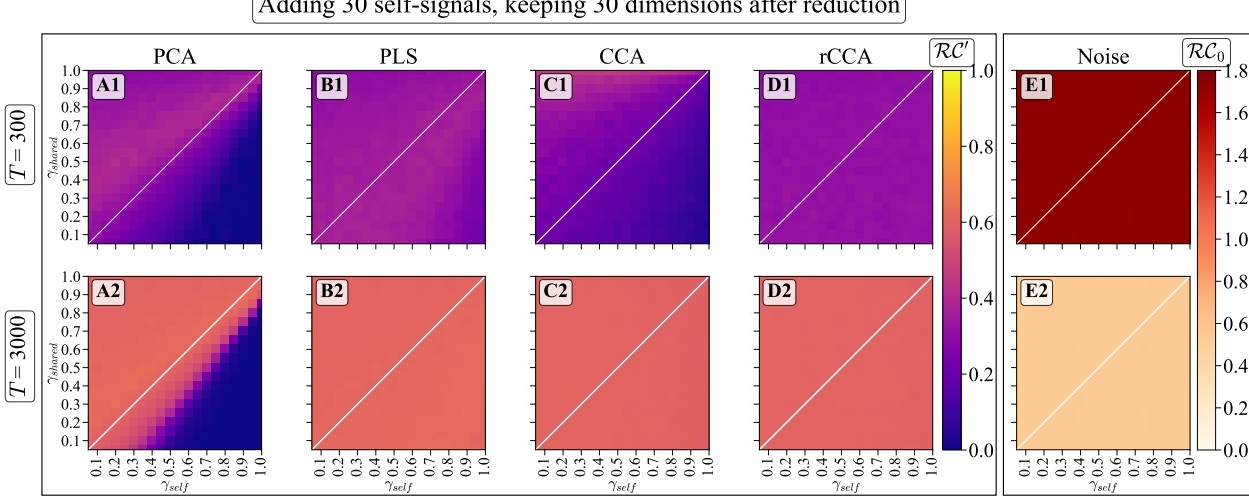

Figure 4: DR performance for $|Z_X| = |Z_Y| = m_{\text{self}} > m_{\text{shared}}$). PCA now detects shared signals even when they are weaker than the self signals. However, the quality of reconstruction is significantly lower than in Fig. 2. PLS detects signals in a larger part of the phase space, but also with a significant reduction in quality, which improves with sampling. CCA has its usual problem for $T \ll N_X$, and, like PLS, it has a significantly lower reconstruction quality than in the regime in Fig. 3. rCCA is able to detect the signal in the whole phase space, but again with worse quality. Finally, spurious correlations are high, though they decrease with better sampling.

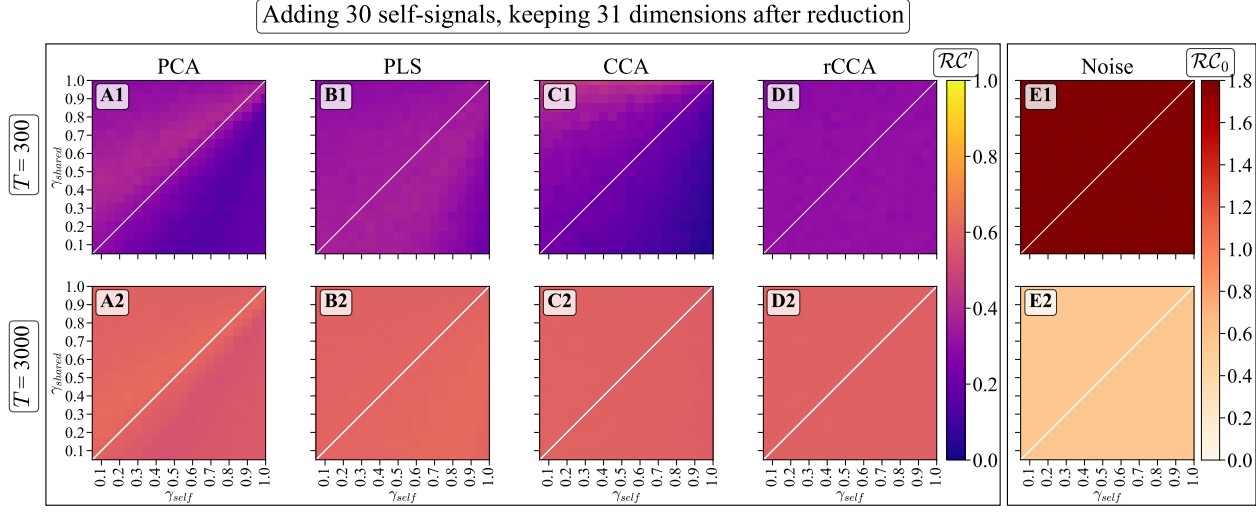

Figure 5: PCA, PLS, CCA, rCCA, and noise results when 31 dimensions are kept after reduction ($|Z_X| = |Z_Y| = m_{\text{self}} + m_{\text{shared}}$). PCA now can detect more shared signals when they are weaker than the self signals (A1), however, with a significantly lower quality compared to Fig. 2, but suddenly explores the whole phase space, still with lower accuracy than Case 1. PLS, CCA, rCCA, and noise show similar behavior to figure 4.

to the success. If $\tilde{q}$ is small, the signal is drowned in the sampling noise, and adding more retained dimensions hurts the DR process. This expresses itself as a peak for SDR performance around $|Z_X| = m_{\text{shared}}$. For IDR, the peak is around $|Z_X| = m_{\text{self}} + m_{\text{shared}}$, thus requiring more data to achieve performance similar to SDR.

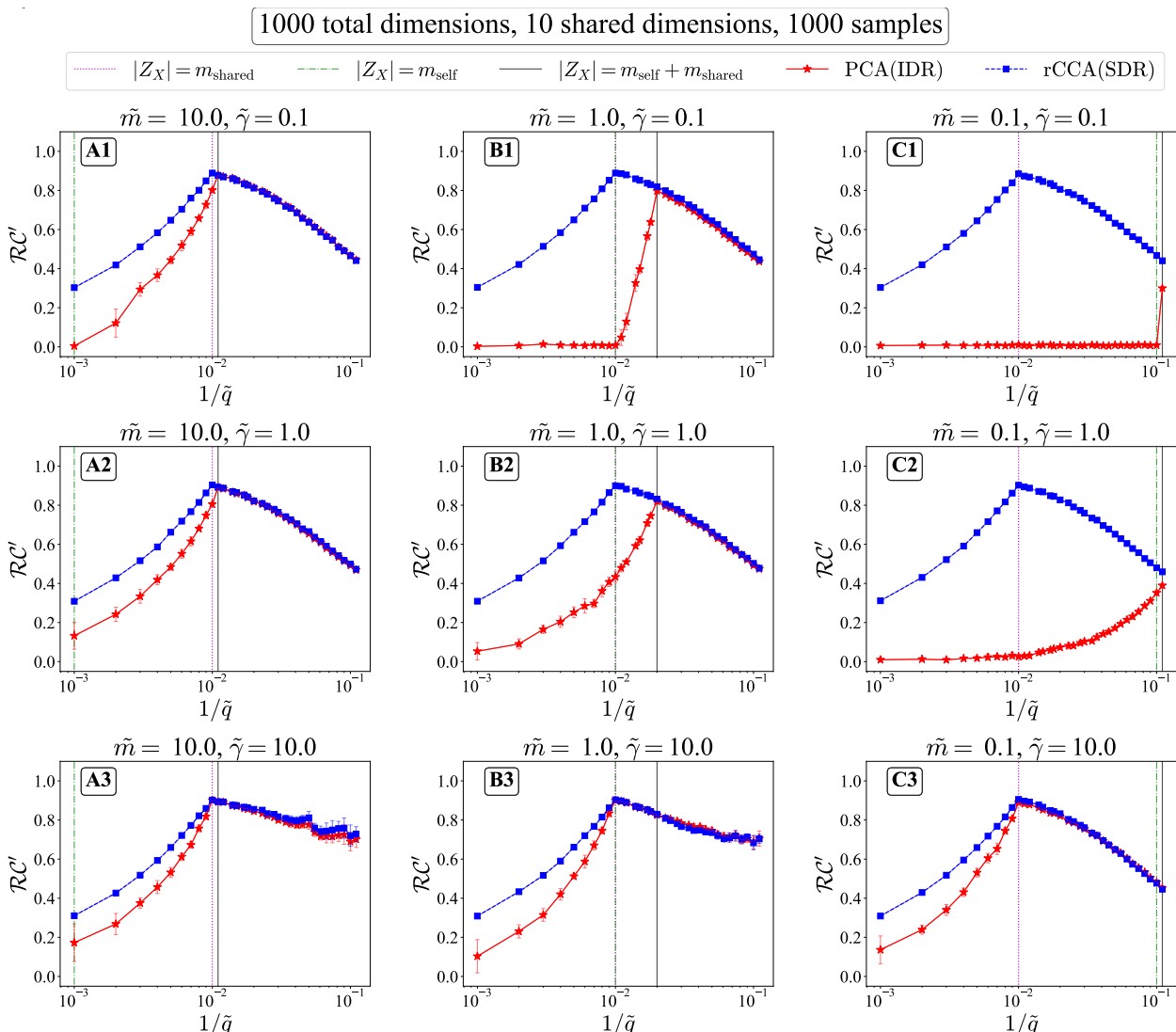

Figure 6: Performance of PCA (IDR) and rCCA (SDR) for different values of the relevant parameters of the model: the number of samples per dimension of the compressed variable $(\tilde{q})$, the strength of shared signals relative to the self ones $(\tilde{\gamma})$, and the ratio of the number of shared to self signal components $(\tilde{m})$, while fixing the number of samples $(T = 1000)$ and the number of shared dimensions $(m_{\text{shared}} = 10)$. Note that increasing $1/\tilde{q}$ (left to right) corresponds to increasing the dimension of the latent space $|Z_X|$ at a fixed number of samples $T$.

These results can be understood from a simple counting argument. In SDR, the objective function is based on the cross-covariance or the cross-correlation matrix. In our linear model, the cross-covariance $C_{XY} \propto \sigma_P^2 Q_X^\top Q_Y$ (up to sampling effects) should depend on the quenched projection matrices $Q_X$ and $Q_Y$. The number of parameters defining these quenched projection matrices is $m_{\text{shared}} \times N_X$ and $m_{\text{shared}} \times N_Y$ respectively. To be able to reconstruct the cross-covariance or the correlation matrix from a DR method, we need a number of observations that scales with this number of parameters. The data matrices $X$ and $Y$ have $T \times N_X$ and $T \times N_Y$ entries respectively. Thus, we need $T \geq m_{\text{shared}}$ samples to have a chance of having enough samples to capture the shared signal. In contrast, in IDR, the objective function is built from the variance matrix $C_{XX} \propto \sigma_{R_X}^2 I + \sigma_{U_X}^2 V_X^\top V_X + \sigma_P^2 Q_X^\top Q_X$, which jointly depends on the quenched projection matrices $V_X$ and $Q_X$, having $m_{\text{self},X} \times N_X$ and $m_{\text{shared}} \times N_X$ parameters, respectively. Therefore, to get enough samples to reconstruct large correlation dimensions in this matrix, we would need at least

$T \geq m_{\text{self},X} + m_{\text{shared}}$ measurements. Furthermore, not all parts of the variance matrix $C_{XX}$ are relevant to the shared signal. Thus, the relative strength of the shared signal to the self signal and the number of self and shared signals will determine which part of the signal is captured first. Finally, adding unnecessary dimensions will hurt performance, as there will be fewer samples per dimension.

We observe that the performance of rCCA (SDR) is almost independent of changing $\tilde{m}$ or $\tilde{\gamma}$, indicating that it focuses on shared dimensions even if the latter is masked by self signals. The algorithm crucially depends on $\tilde{q}$, where adding more dimensions (decreasing $\tilde{q}$) than needed hurts the reduction. This is because, for a fixed number of samples, the reconstruction of each dimension then gets worse. In contrast, for PCA (IDR), the performance depends on all three relevant parameters, $\tilde{q}$, $\tilde{m}$, and $\tilde{\gamma}$. At some parameter combinations, the performance of IDR in reconstructing shared signals approaches SDR. However, in all cases, SDR never performs worse than IDR on this task.

### 4.2   Beyond The Linear Model: Noisy MNIST

### 4.2.1   The Dataset

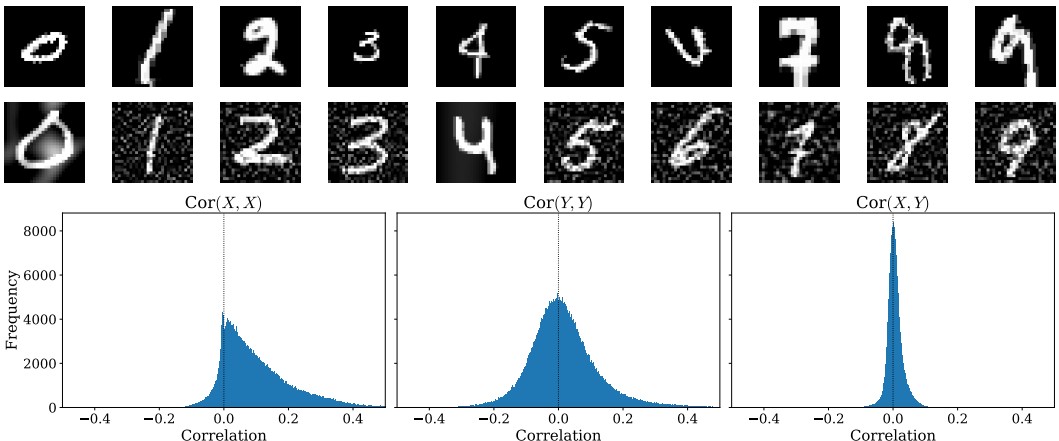

Figure 7: Dataset containing paired MNIST digit samples sharing only the same identity *(shared signal)*. The first row $(X)$ shows MNIST digits randomly subjected to scaling, $(0.5 - 1.5)$, and rotation with an angle of $(0 - \pi/2)$, while the second row $(Y)$ shows MNIST digits with an added background Perlin noise *(self signals)*. In the bottom row, histograms of self correlations for the $X$ and $Y$ datasets (left and middle, respectively) illustrate a wide range of correlations, while the histogram of the cross correlation between $X$ and $Y$ (right) demonstrates a smaller range.

To analyze linear DR methods on nonlinear data, we followed the same procedure as in Fig. 6 for a dataset inspired by the noisy MNIST dataset (LeCun et al., 1998; Wang et al., 2015; 2016; Abdelaleem et al., 2023). This dataset has two distinct views of data, each of dimensionality $28 \times 28$ pixels, examples of which are shown in Fig. 7. The first view is an image of the digit subjected to a random rotation within an angle uniformly sampled between 0 and $\frac{\pi}{2}$, along with scaling by a factor uniformly distributed between 0.5 and 1.5. The second view consists of another image with the same digit identity with an additional background layer of Perlin noise (Perlin, 1985), with the noise factor uniformly distributed between 0 and 1. Both views are normalized to an intensity range of $[0, 1)$, then flattened to form an array of 784 dimensions.

To cast this dataset into our language, we shuffled the images within labels, retaining the shared label identity (that is the shared signal), but we still have the view-specific details (which is the self signal). This resulted in a total dataset size of $\sim 56k$ images for training and $\sim 7k$ images for testing. The correlation histogram of $X$ (or $Y$) with itself shows a relatively wide spectrum when compared to the cross correlation between $X$ and $Y$, highlighting that the self signal is stronger, and can lead to different DR methods overseeing the shared one. The complexity of the tasks makes it sufficiently challenging, serving as a good benchmark for evaluating the performance of the different DR techniques.

### 4.2.2 Results

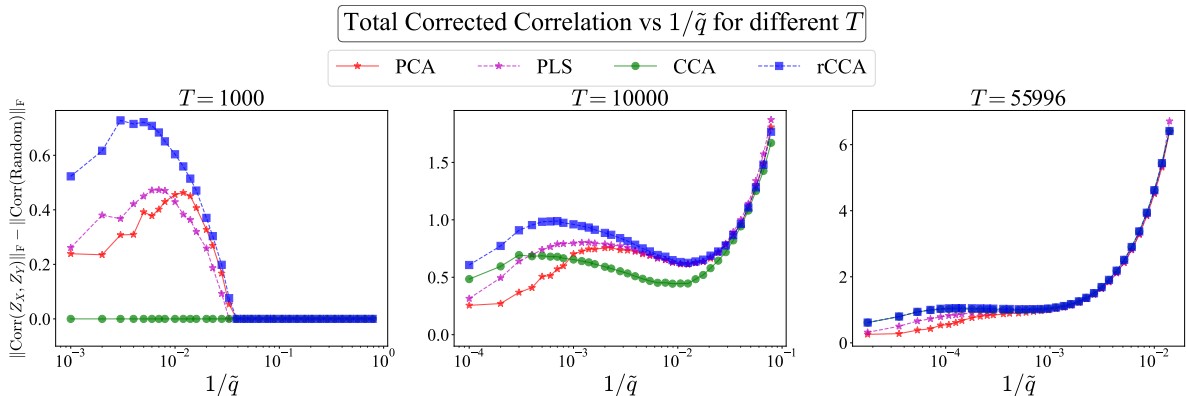

Figure 8: Performance of PCA, PLS, CCA, rCCA applied to the modified Noisy MNIST dataset across varying sampling scenarios. Each panel represents different sample sizes (1000, 10,000, and approximately 56,000 samples). The x-axis denotes the inverse of the number of samples per retained dimensions ($1/\tilde{q}$), while the y-axis represents the total corrected correlation between the obtained low-dimensional representations $Z_X$ and $Z_Y$.

Figure 8 shows the performance of PCA, PLS, CCA, and rCCA applied to the modified Noisy MNIST dataset for varying sampling scenarios. The three panels are evaluated for different sample sizes (1000, 10,000, and $\sim 56,000$ samples), from undersampled to the full dataset.

In each scenario, the training samples are used for the DR methods. Subsequently, the learned projection matrices onto the singular directions are used to transform a separate test dataset of around 7,000 samples into low-dimensional spaces, yielding $Z_X$ and $Z_Y$. The correlation between these transformed spaces is computed using the Frobenius norm of the correlation matrix. As before, we then subtracted from it the correlation value obtained from a random matrix of the same size. This difference is then plotted against $1/\tilde{q}$, which is the measure of how many dimensions are retained at each sampling ratio.

In the undersampled scenario (1000 samples), rCCA and PLS demonstrate an early detection (in terms of the number of kept dimensions after reduction) of shared signals, whereas PCA initially lags behind. As the number of dimensions increases, all methods exhibit a decline in correlation due to increased noise as we have fewer samples per dimension. CCA does not work in this scenario, since covariance matrices are degenerate.

Upon increasing the sample size (10,000 samples), a similar pattern emerges initially, where all methods experience an increase in total correlation till a certain number of kept dimensions is reached, then a decline when adding more dimensions. The decline is because one needs to estimate more singular vectors from the same number of samples. However, beyond a certain number of singular vectors, an increase in correlation is observed. This is because the number of vectors is now sufficient to learn both the shared and the self signals. We observe that rCCA maintains superior performance, while PCA reaches peak correlation at a higher number of kept dimensions, providing a rough estimation of the number of true self and shared signals. With the full dataset (approximately 56,000 samples), a similar trend is seen. Yet CCA's performance approaches that of rCCA.

Notably, these analyses confirm that linear Simultaneous Dimensionality Reduction (SDR) are consistently better at detecting shared signals than linear Independent Dimensionality Reduction (IDR), even in some nonlinear datasets.

## 5    Discussion

We used a generative linear model which captures multiple desired features of multimodal data with shared and non-shared signals. The model focused only on data with two measured modalities. However, while not a part of this study, the model can be readily extended to accommodate more than two modalities (e. g., $X_i = R_i + U_i V_i + P Q_i$ for $i = 1, ..., n$, where $n$ represents the number of modalities). Then, methods such as Tensor CCA, which can handle more than two modalities (Luo et al., 2015), can be used to get insight into DR on such data.

We analyzed different DR methods on data from this model in different parameter regimes. Linear SDR methods were clearly superior to their IDR counterparts for detecting shared signals. In particular, SDR performance peaked around $|Z_X| = m_{\mathrm{shared}}$ and IDR performance peaked around $|Z_X| = m_{\mathrm{self}} + m_{\mathrm{shared}}$. Thus, SDR required fewer samples for similar detection performance to IDR. We observed similar results on a nonlinear dataset as well. We thus make a strong practical suggestion that, whenever the goal is to reconstruct a low dimensional representation of covariation between two components of the data, IDR methods (PCA) should always be avoided in favor of SDR. Of the examined SDR approaches, rCCA is a clear winner in all parameter regimes and should always be preferred.

While we performed the analysis using specific examples of SDR, such as PLS, (r)CCA, and specific examples of IDR, like PCA, we anticipate similar results to hold for other SDR and IDR methods. For instance, methods that optimize a certain aspect within one modality (e. g., Independent Component Analysis Hyvärinen & Oja (2000), Nonnegative Matrix Factorization Lee & Seung (2000), Autoencoders Hinton & Salakhutdinov (2006)), or methods using multiple modalities but retaining only certain aspects within each modality (e. g., Multiview PCA Xia et al. (2021)) should be able to detect the shared signals only if they are stronger than the self signals, given the other detection criteria (of having enough number of kept dimensions and enough samples to sample them properly) are met. Otherwise, they will likely detect the self signals first, wasting some of their statistical power. Alternatively, methods that work across modalities (e. g., Cross-modal Factor Analysis Li et al. (2003), Deep Variational Symmetric Information Bottleneck Abdelaleem et al. (2023)) should identify the shared signal first. We leave the verification of this hypothesis across a broader range of methods for future work. To analyze the behavior of such more complicated methods, we will need generative models of data that have low-dimensional structure, which cannot, nonetheless, be approximated by methods that rely on singular value spectra and their nonlinear generalizations; first steps towards creating such generative models were recently taken Fleig & Nemenman (2022b).

These findings explain the results of, for example, Sani et al. (2021) and others that SDR can recover joint neuro-behavioral latent spaces with fewer latent dimensions and using fewer samples than IDR methods. Further, our observation that SDR is always superior to IDR in the context of our model corroborates the theoretical findings of Martini & Nemenman (2024), who proved a similar result in the context of discrete data and a different SDR algorithm, namely the Symmetric Information Bottleneck (Friedman et al., 2013). Vogelstein et al. (2021) made similar conclusions using conditional covariance matrices for the reduction in the context of classification. More recent work of Abdelaleem et al. (2023) showed similar results using deep variational methods. Collectively, these diverse investigations, linear and nonlinear, theoretical, computational, and empirical, provide strong evidence that generic (not just linear) SDR methods are likely to be more efficient in extracting covariation than their IDR analogs.

Our study also answers an open question in the literature surrounding the effectiveness of SDR techniques. Specifically, there has been debate about whether PLS, an SDR method, is effective at low sampling (Chin & Newsted, 1999; Hair et al., 2011; Goodhue et al., 2006; 2012). Our results show that SDR is not necessarily effective in the undersampled regime. It works well when the number of samples per retained dimension is high (even if the number of samples per observed dimension is low), but only when the dimensionality of the reduced description is matched to the actual dimensionality of the shared signals.

Finally, our results can be used as a diagnostic test to determine the number of shared versus self signals in data. As demonstrated in Fig. 6, total correlations between $Z_X$ and $Z_Y$ obtained by applying PCA and rCCA increase monotonically as the dimensionality of $Z$s increases, until this dimensionality becomes larger than the signal dimensionality. For PCA, the signal dimensionality is equal to the sum of the number of

the shared and the self signals, $m_{\text{shared}} + m_{\text{self}}$. For rCCA, it is only the number of the shared signal. Thus increasing the dimensionality of the compressed variables and tracking the performance of rCCA and PCA until they diverge can be used to identify the number of self signals in the data, provided that the data, indeed, has a low-dimensional latent structure. This approach can be a valuable tool in various applications, where the characterization of shared and self signals in complex systems can provide insights into their structure and function.

In summary, we highlight a general principle that, when searching for a shared signal between different modalities of data, SDR methods are preferable to IDR methods. Additionally, the differences in performance between the two classes of methods can tell us a lot about the underlying structure of the data. Finally, for a limited number of samples, naive approaches, such as increasing the number of compressed dimensions indefinitely to overcome the masking of shared signals by self signals are infeasible. Thus, the use of SDR methods becomes even more essential in such cases.

## 6  Limitations and Future Work

While this work has provided useful insight, the assumptions made here may not fully capture the complexity of real-world data. Specifically, our data is generated by a linear model with random Gaussian features. It is unlikely that real data have this exact structure. Therefore, there is a need for further exploration of the advantages and limitations of linear DR methods on data that have a low-dimensional, but nonlinear shared structure. This can be done using more complex nonlinear generative models, such as nonlinearly transforming the data generated by Eq. (1-2), or random feature two-layered neural network models (Rocks & Mehta, 2022). Alternatively, analyzing the model, Eq. (1) using various theoretical techniques (Borga et al., 1997; Vogelstein et al., 2021; Potters & Bouchaud, 2020) is likely to offer even more insights into its properties. Collectively, these diverse approaches would aid our understanding of different DR methods under diverse conditions.

A different possible future research direction is to explore the performance of nonlinear DR methods on data from generative models with a latent low-dimensional nonlinear structure. Autoencoders and their variational extensions are a natural extension of IDR to learn nonlinear reduced dimensional representations (Hinton & Salakhutdinov, 2006; Kingma & Welling, 2014; Higgins et al., 2016). Meanwhile, Deep CCA and its variational extensions (Andrew et al., 2013; Wang et al., 2015; Chandar et al., 2016; Wang et al., 2016) should be explored as a nonlinear version of SDR. Both of these types of methods can potentially capture more complex relationships between the modalities and improve the quality of the reduced representations, and while recent work suggests that (Abdelaleem et al., 2023), it is not clear if the SDR class of methods is always more efficient than the IDR one.

Further, our analysis depends on the choice of metric used to quantify the performance of DR, and different choices should also be explored. For example, to capture nonlinear correlations, mutual information can be utilized to quantify the relationships between the reduced representations.

Despite the aforementioned limitations, we believe that our work provides a compelling addition to the body of knowledge that SDR outperforms IDR in detecting shared signals quite generally.

### Acknowledgments

We thank Sean Ridout for providing feedback on the manuscript, IN thanks Philipp Fleig for useful discussions. This work was funded, in part, by NSF Grant No. 2010524 and 2014173 and by the Simons Investigator award to IN.

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

# A   Appendix

## A.1   Linear Dimensionality Reduction Methods

### A.1.1   Principal Components Analysis (PCA)

PCA is a widely used linear IDR method that aims to find the orthogonal principal directions, such that a few of them explain the largest possible fraction of the variance within the data. PCA decomposes the covariance matrix of the data matrix $X$, $C_{XX} = \frac{1}{T} X^\top X$, into its eigenvectors and eigenvalues through singular value decomposition (SVD). The SVD yields orthogonal directions, represented by the vectors $w_X^{(i)}$, that capture the most significant variability in the data. In most numerical implementations (Pedregosa et al., 2011), these directions are obtained consecutively, one by one, such that the dot product between

any two directions is zero $w_X^{(i)} \cdot w_X^{(j)} = \delta_{ij}$. The eigenvectors $w_X^{(i)}$ are obtained as the best solution to the optimization problem:

$$w_X^{*(i)} = \arg\max_{w_X^{(i)}} \frac{w_X^{(i)\top} X^{(i)\top} X^{(i)} w_X^{(i)}}{w_X^{(i)\top} w_X^{(i)}}. \tag{6}$$

Here $X^{(i)}$ is the $i$th deflated matrix where $X^{(1)}$ is the original matrix, and for every subsequent $i + 1$, the matrix is deflated by subtracting the projection of $X$ on the obtained weights: $X^{(i+1)} = X - \Sigma_{s=1}^i X w_{(s)} w_{(s)}^\top$. The eigenvectors are sorted in decreasing order according to their corresponding eigenvalues, and the first $k$ eigenvectors $w_X^{(i=1:k)}$ are selected to form the projection matrix $W_X$. The obtained vectors determine the size of the reduced form $Z_X$, where $|Z_X| = k$ is the number of vectors retained from the decomposition of $X$. The vectors $w_X^{(i)}$ are then stacked together to form the projection matrix $W_X$. The low-dimensional representation $Z_X$ is then obtained by multiplying the original data matrix $X$ with this projection matrix, resulting in the reduced data matrix $Z_X = X W_X$. Similar treatment is done for $Y$ in order to obtain $Z_Y = Y W_Y$

One of the main advantages of PCA is its simplicity and efficiency. However, one of the drawbacks of this method is that it performs DR for $X$ and $Y$ independently, and one then searches for relations between $Z_X$ and $Z_Y$ by regressing one on the other—the so-called *Principal Components Regression*. Thus obtained low-dimensional descriptions may capture variance but not the covariance between the two datasets.

### A.1.2  Partial Least Squares (PLS)

PLS performs SDR by finding the shared signals that explain the maximum covariance between two sets of data (Wold et al., 2001). PLS performs the SVD of the covariance matrix $C_{XY} = \frac{1}{T} X^\top Y$ (or equivalently $C_{YX} = \frac{1}{T} Y^\top X$). The left and right singular vectors $(w_X^{*(i)}, w_Y^{*(i)})$ are obtained consecutively pair by pair such that $w_X^{(i)} \cdot w_Y^{(j)} = \delta_{ij}$. They are solutions of the optimization problem:

$$(w_X^{*(i)}, w_Y^{*(i)}) = \arg\max_{w_X^{(i)}, w_Y^{(i)}} \frac{w_X^{(i)\top} X^{(i)\top} Y^{(i)} w_Y^{(i)}}{\sqrt{(w_X^{(i)\top} w_X^{(i)})(w_Y^{(i)\top} w_Y^{(i)})}} \tag{7}$$

The matrices $X^{(i)}, Y^{(i)}$ are deflated in a similar manner to PCA A.1.1[6]. The singular vectors are sorted in the decreasing order of their corresponding singular values, and the first $k$ vectors are selected to form the projection matrices $(W_X, W_Y)$. The obtained vectors determine the size of the reduced form $(Z_X, Z_Y)$, where $|Z_X| = |Z_Y| = k$ is the number of vectors retained. The vectors $(w_X^{(i)}, w_Y^{(i)})$ are then stacked together to form the projection matrices $(W_X, W_Y)$ respectively. The low-dimensional representations $(Z_X, Z_Y)$ are obtained by projecting the original data matrices $(X, Y)$ onto these projection matrices: $Z_X = X W_X$, and $Z_Y = Y W_Y$.

In summary, PLS performs simultaneous reduction on both datasets, maximizing the covariance between the reduced representations $Z_X$ and $Z_Y$. This property makes PLS a powerful tool for studying the relationships between two datasets and identifying the underlying factors that explain their joint variability. In practice, PLS can suffer from widely distinct variances along different singular directions, as well as covariances within $X$ or $Y$. At the very least, in practical implementations, each component of $X$ and $Y$ is typically standardized to unit variance before applying PLS to avoid some (but not all) of these issues.

### A.1.3  Canonical Correlations Analysis (CCA)

CCA is another SDR method, which aims to find the directions that explain the maximum correlation between two datasets Hotelling (1936). However, unlike PLS, CCA obtains the shared signals by performing SVD on the correlation matrix $\frac{C_{XY}}{\sqrt{C_{XX}} \sqrt{C_{YY}}}$. The singular vectors $(w_X^{*(i)}, w_Y^{*(i)})$ are obtained consecutively

---

[6]This PLS implementation with iterative deflation is often referred to as `PLSCanonical`. An alternative simpler approach based on direct SVD calculation of the covariance matrix is often known as `PLSSVD` Pedregosa et al. (2011).

pair by pair such that $w_X^{(i)} \cdot w_Y^{(j)} = \delta_{ij}$. CCA enforces the orthogonality of $w_X^{(i)}, w_Y^{(i)}$ independently as well, such that $w_X^{(i)} \cdot w_X^{(j)} = w_Y^{(i)} \cdot w_Y^{(j)} = \delta_{ij}$. The singular vectors are obtained by solving the optimization problem:

$$(w_X^{*(i)}, w_Y^{*(i)}) = \underset{w_X^{(i)}, w_Y^{(i)}}{\arg\max} \frac{w_X^{(i)\top} X^{(i)\top} Y^{(i)} w_Y^{(i)}}{\sqrt{(w_X^{(i)\top} X^{(i)\top} X^{(i)} w_X^{(i)})(w_Y^{(i)\top} Y^{(i)\top} Y^{(i)} w_Y^{(i)})}}. \tag{8}$$

Like in PLS A.1.2, the matrices $X^{(i)}, Y^{(i)}$ are deflated in a similar manner. In addition, the first $k$ singular vectors $(w_X^{*(i)}, w_Y^{*(i)})$ are stacked together to form the projection matrices $(W_X, W_Y)$, which then are used to obtain the reduced data matrices $Z_X = XW_X$, and $Z_Y = YW_Y$.

One of the key differences between PLS and CCA is that while both perform SDR, CCA also simultaneously performs IDR implicitly. Indeed, it involves multiplication of $C_{XY}$ by $C_{XX}^{-1/2}$ on the left and $C_{YY}^{-1/2}$ on the right, which, in turn, requires finding singular values of the $X$ and the $Y$ data matrices independently.

### A.1.4 Regularized CCA - rCCA

While CCA is a useful method for finding the maximum correlating features between two sets of data, it does have some limitations. Specifically, in the undersampled regime, where $T \leq \max(N_X, N_Y)$, the matrices $C_{XX}$ and $C_{YY}$ are singular and their inverses do not exist. Using the pseudoinverse to solve the problem can lead to numerical instability and sensitivity to noise. Regularized CCA (rCCA) (Vinod, 1976; Årup Nielsen et al., 1998) overcomes this problem by adding a small regularization term to the covariance matrices, allowing them to be invertible. Specifically, one tales

$$\tilde{C}_{XX} = C_{XX} + c_X I_X, \tag{9}$$

$$\tilde{C}_{YY} = C_{YY} + c_Y I_Y, \tag{10}$$

where $\tilde{C}_{XX}, \tilde{C}_{YY}$ are the new regularized matrices, $c_X, c_Y > 0$ are small regularization parameters and $I_X, I_Y$ are identity matrices with sizes $N_X \times N_X, N_Y \times N_Y$ respectively.

This original implementation of rCCA resulted in correlation matrices with diagonals not equal to one. Thus, a better implementation uses a different form of regularization (Årup Nielsen et al., 1998) by adding the regularization parameters $c_X$ and $c_Y$ individually to the equations as an affine combination (i. e., $\sum_i^n c_i = 1$) as the following:

$$\tilde{C}_{XX} = \frac{1}{T}(c_{X_1} w_X^\top X^\top X w_X + c_{X_2} w_X^\top w_X), \tag{11}$$

$$\tilde{C}_{YY} = \frac{1}{T}(c_{Y_1} w_Y^\top Y^\top Y w_Y + c_{Y_2} w_Y^\top w_Y). \tag{12}$$

This results in the regularized equations for $X$ and $Y$:

$$\tilde{C}_{XX} = \frac{1}{T}\big((1-c_X) w_X^\top X^\top X w_X + c_X w_X^\top w_X\big), \tag{13}$$

$$\tilde{C}_{YY} = \frac{1}{T}\big((1-c_Y) w_Y^\top Y^\top Y w_Y + c_Y w_Y^\top w_Y\big), \tag{14}$$

where $c_X$ and $c_Y$ are the regularization parameters, with values between 0 and 1. Hence our new optimization problem becomes:

$$(w_X^{*(i)}, w_Y^{*(i)}) = \underset{w_X^{(i)}, w_Y^{(i)}}{\arg\max} \frac{w_X^{(i)\top} X^{(i)\top} Y^{(i)} w_Y^{(i)}}{\sqrt{\tilde{C}_{XX} \tilde{C}_{YY}}}. \tag{15}$$

Writing the regularization conditions in this form is, in fact, a convex interpolation problem between PLS and CCA, which is a more robust solution and does not suffer from shortening the length of correlations due to the added regularization. As a result, this implementation of rCCA achieves the best accuracy among all other methods [7].

---

[7] Conceptually, the main difference between PLS and CCA is that PLS does not enforce orthogonality among the weights $w_X^{(i)}$ and $w_Y^{(i)}$ that diagonalize $C_{XX}$ and $C_{YY}$, whereas CCA does. For instance, while two pairs of singular vectors $(w_X^{(1)}, w_Y^{(1)})$ and

## A.2 Assessing Success and Sampling Noise Treatment

To assess the success of DR, we calculated the ratio between the total correlation between $Z_{X_{\text{test}}}$ and $Z_{Y_{\text{test}}}$, defined as in Eq. (5), and the total correlation between $X$ and $Y$, which we input into the model. Specifically, we take the total correlation as the Frobenius norm of the correlation matrix, $||A||_F = \sqrt{\sum_i \sigma_i^2(A)}$, where $\sigma(A)$ are the singular values of the matrix $A$. Therefore, the metric of the quality of the DR is

$$\mathcal{RC} = \frac{||\text{Corr}(Z_{X_{\text{test}}}, Z_{Y_{\text{test}}})||_F}{||\text{Corr}(P, P)||_F} = \frac{||\text{Corr}(Z_{X_{\text{test}}}, Z_{Y_{\text{test}}})||_F}{m_{\text{shared}}}, \tag{16}$$

where Corr stands for the correlation matrix between its arguments, and we use $||\text{Corr}(P, P)||_F = m_{\text{shared}}$ as the total shared correlation that one needs to recover. Statistical fluctuations aside, $\mathcal{RC}$ should vary between zero (bad reconstruction of the shared variables) and one (perfect reconstruction)[8].

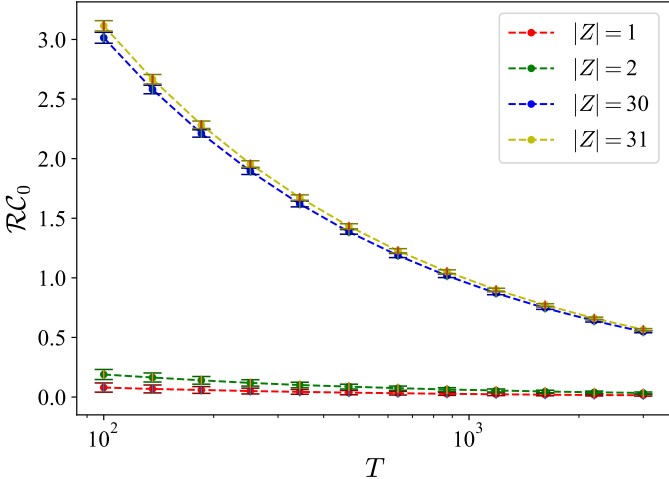

Figure 9: The resulting correlations are averages of all the points in the phasespace, then averaged over 10 different realizations of the matrices. The error bars are for two standard deviations around the mean

In many real-world applications, the number of available samples, $T$, is often limited compared to the dimensionality of the data, $N_X$ and $N_Y$. This undersampling can introduce spurious correlations. We are not aware of analytical results to calculate the effects of the sampling noise on estimating singular values

---

$(w_X^{(2)}, w_Y^{(2)})$ are mutually orthogonal as pairs, $w_X^{(1)}$ and $w_X^{(2)}$ are not orthogonal to each other. This partial overlap can result in signals not being fully expressed in these directions, reducing the total correlation compared to CCA, where each direction $w_X^{(i)}$ and $w_Y^{(i)}$ is orthogonal to every other direction, allowing each signal to fully occupy these directions and maximizing total correlation. Therefore, our goal is to make CCA work, but due to inversion issues, rCCA serves as a substitute. This is evident in Figures 1-5, where, under good sampling conditions, rCCA and CCA yield almost identical results.

[8]We note that the choice of $\mathcal{RC}$ is not unique, and other choices could be valid as well. However, we believe that $\mathcal{RC}$ is suitable for our study. For example, one could use mutual information $I(Z_X; Z_Y)$ Shannon (1948) to quantify how much information $Z_X$ and $Z_Y$ carry about each other. Mutual information is a fundamental statistic that is 0 if and only if the two variables are statistically independent. However, its estimation is a challenging problem, as it depends on the probabilities $p(Z_X)$, $p(Z_Y)$, and $p(Z_X, Z_Y)$ Antos & Kontoyiannis (2001); Paninski (2003). In this setup, while the individual matrices of $X$ and $Y$ are Gaussian with known probability density functions, their product is not. Consequently, a closed analytical form of mutual information is not known. Even if we assumed $Z_X$ and $Z_Y$ to be Gaussian and used the formula $I(Z_X; Z_Y) = -\frac{1}{2} \ln |C_{X,Y}|$, where $|.|$ denotes the determinant of the original correlation matrix $C_{XY}/\sqrt{C_{XX} \cdot C_{YY}}$, we might end up with directions that do not carry information due to having more dimensions in $Z_X$ (or $Z_Y$) than needed, thus creating numerical instabilities. These instabilities are often addressed by employing a threshold on the determinant (considering the determinant as the product of singular values, we threshold the small singular values). However, we chose not to use such a method, focusing instead on $\mathcal{RC}$ for the following reasons: (i) With this choice, we do not need to employ thresholding, and by using a metric that uses the singular values of the correlation matrix in an additive fashion, rather than multiplicative as in mutual information, we avoid this issue. (ii) Correlation is an intuitive metric, commonly used by practitioners, with clear meaning and bounds between -1 and +1. While one could use the covariance matrix rather than the correlation matrix, such a choice does not affect the optimization process, as $X$ and $Y$ are standardized. Therefore, the covariance and correlation matrices of the original data are equivalent, and the difference would lie in the calculation of $Cov(Z_X, Z_Y)$ vs $Corr(Z_X, Z_Y)$, potentially increasing the accuracy of PLS in such a case. However, due to the intuitiveness and utility of correlation, we use it in the $\mathcal{RC}$ metric.

in the model in Eq. (1) (Bun et al., 2017). Thus, to estimate the effect of the sampling noise, we adopt an empirical approach. Specifically, we generate two random matrices, $Z_{X_{\text{random}}}$ and $Z_{Y_{\text{random}}}$, of sizes $T \times |Z_X|$ and $T \times |Z_Y|$, respectively. We then calculate the correlation between these matrices, denoted as $\mathcal{RC}_0$, for multiple such trials using the metric in Eq. (16). For random $Z_{X_{\text{random}}}$ and $Z_{Y_{\text{random}}}$, $\mathcal{RC}$ should be zero. However, Fig. 9 shows that, especially for large dimensionalities of the compressed variables and small $T$, the sampling noise results in a significant spurious $\mathcal{RC}_0 > 0$, which may even be larger than 1! Crucially, $\mathcal{RC}_0$ does not fluctuate around its mean across trials, so that the sampling bias is narrowly distributed.

To compensate for this sampling bias, we subtract it from the reconstruction quality metric,

$$\mathcal{RC}' = \mathcal{RC} - \mathcal{RC}_0. \tag{17}$$

It is this $\mathcal{RC}'$ that we plot in all Figures in this paper as the ultimate metric of the reconstruction quality. While subtracting the bias is not the most rigorous mathematically, it provides a practical approach for reducing the effects of the sampling noise.

### A.3 Implementation

We used Python and the `scikit-learn` (Pedregosa et al., 2011) library for performing PCA, PLS, and CCA, while the `cca-zoo` (Chapman & Wang, 2021) library was used for rCCA. For PCA, SVD was performed with default parameters. For PLS, the PLS Canonical method was used with the NIPALS algorithm. For both PLS and CCA, the tolerance was set to $10^{-4}$ with a maximum convergence limit of 5000 iterations. For rCCA, regularization parameters were set as $c_1 = c_2 = 0.1$. All other parameters not explicitly here were set to their default values.

All figures shown in this paper were averaged over 10 independent realizations of $R_X, R_Y, U_X, U_Y, P$, while fixing the projection matrices $V_X, V_Y, Q_X, Q_Y$. We then performed an additional round of averaging everything over 10 realizations of the projection matrices themselves. The simulations were parallelized and run on Amazon Web Services (AWS) servers of various instance types.

### A.4 Extended Figures

In this section, we provide results of simulations similar to the main text Figs. 1, 2, 3, 4, 5, but with a wider range of $T$.

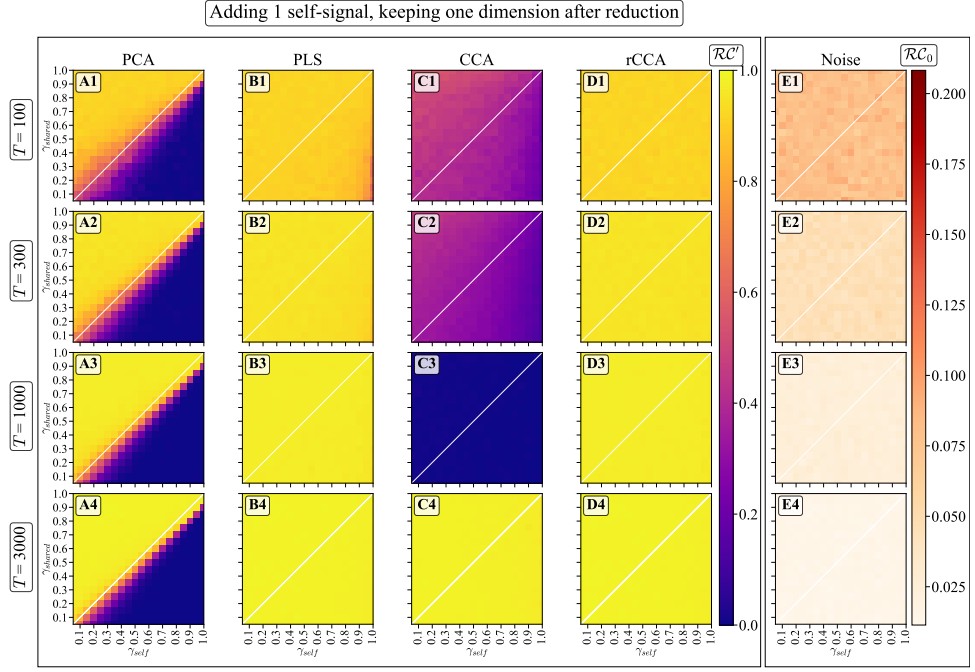

Figure 10: Performance of PCA, PLS, CCA, and rCCA in detecting shared signals with one self signal and one dimension kept after DR.

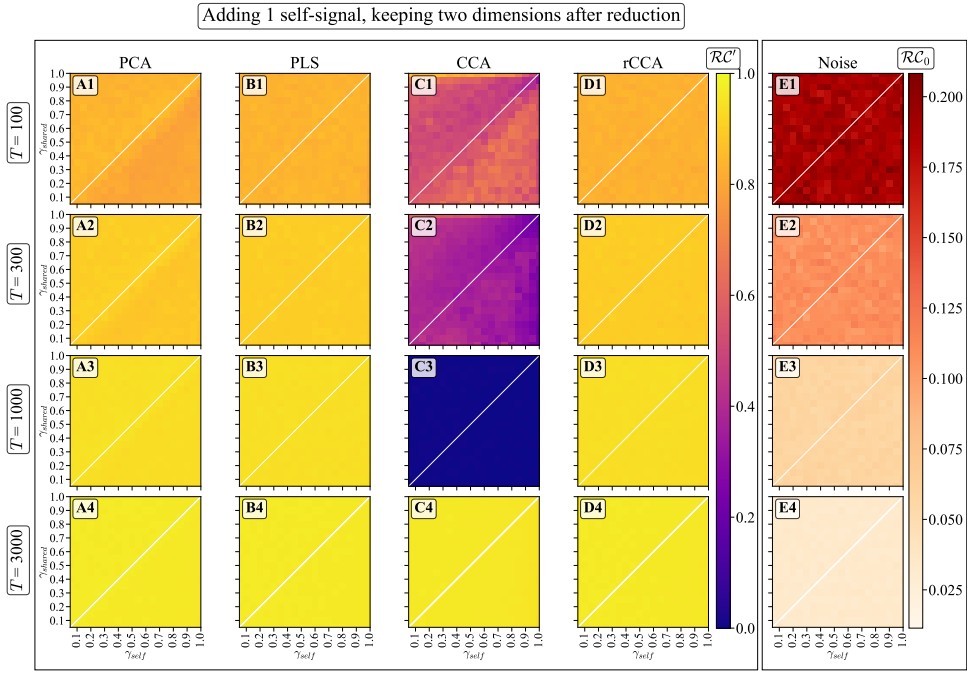

Figure 11: Performance of PCA, PLS, CCA, and rCCA in detecting shared signals with one self signal and two dimensions kept after DR.

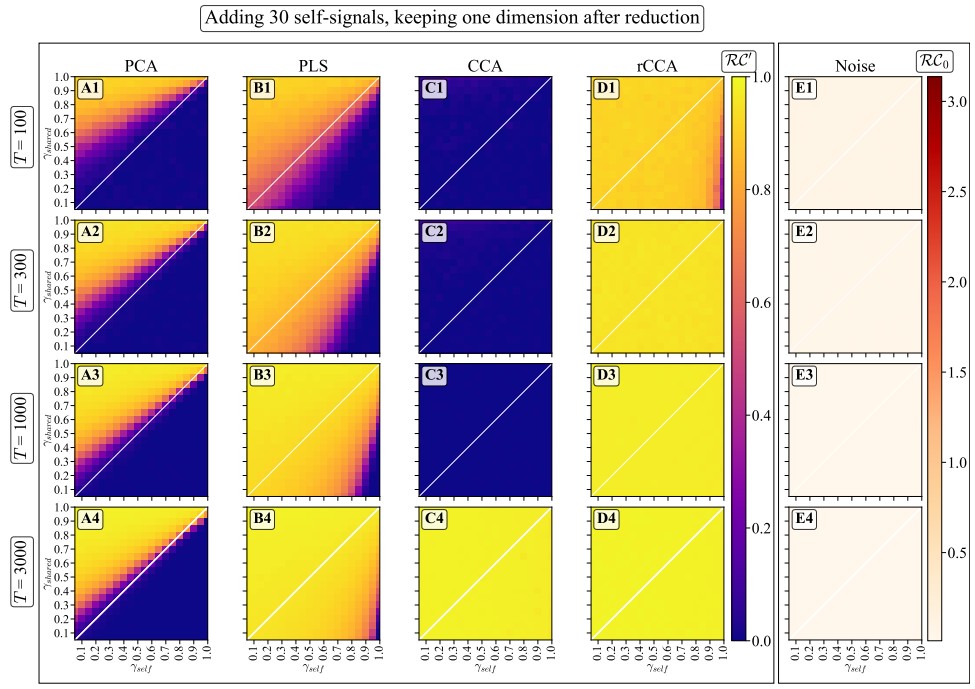

Figure 12: Performance of PCA, PLS, CCA, and rCCA in detecting shared signals with 30 self signals and one dimension kept after DR.

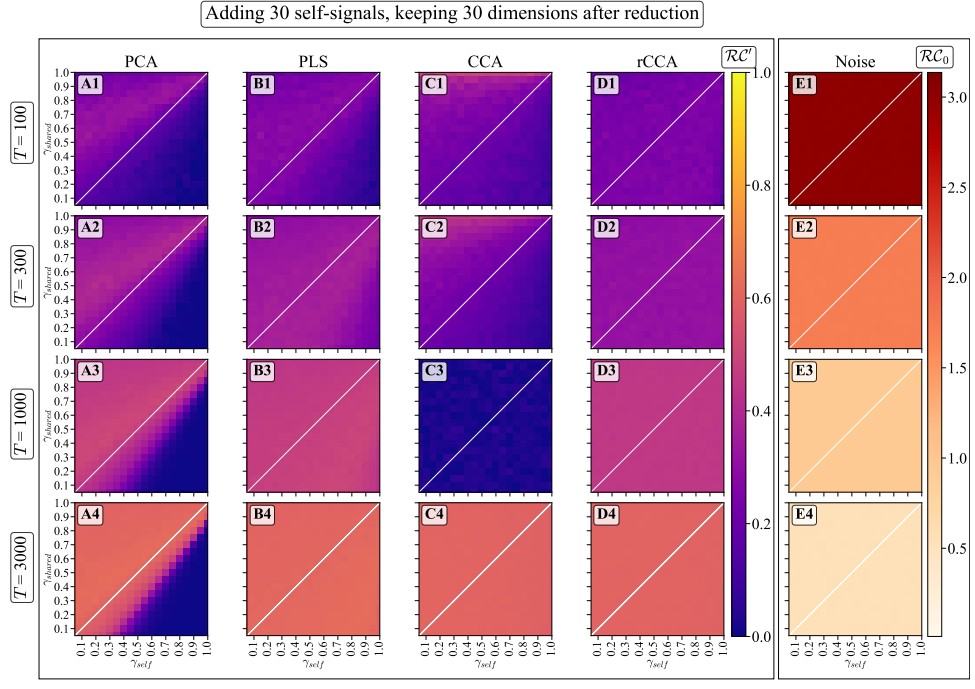

Figure 13: Performance of PCA, PLS, CCA, and rCCA in detecting shared signals among 30 self signals and with 30 dimensions kept after DR.

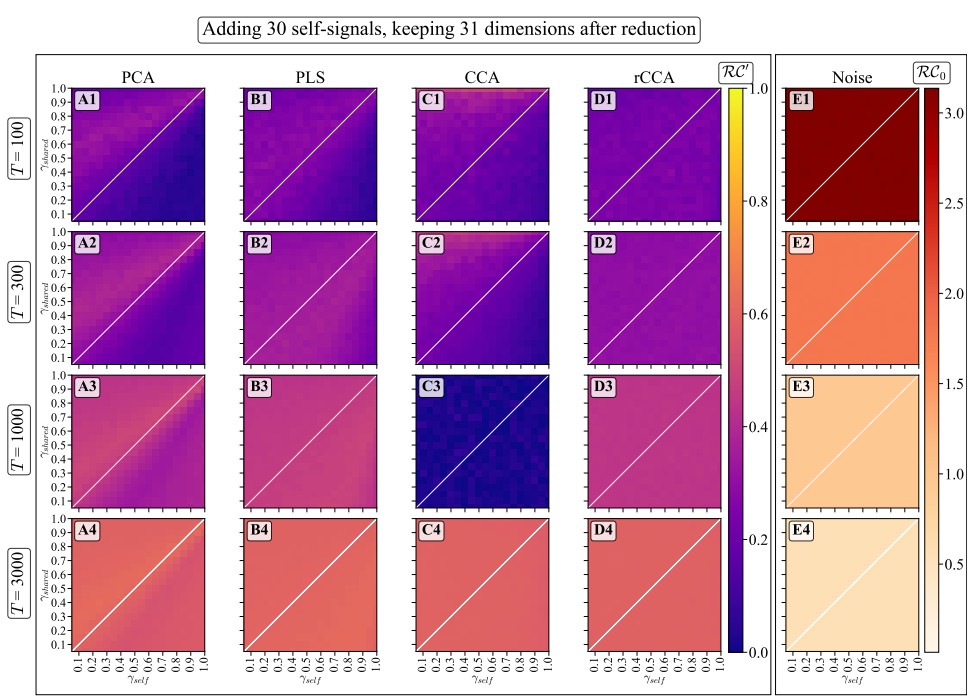

Figure 14: Performance of PCA, PLS, CCA, and rCCA in detecting shared signals among 30 self signals and with 31 dimensions kept after DR.

## A.5  Additional Figures

Here we provide additional figures that show the singular value spectra of the matrices $C_{XX}$ and $C_{XY}$ (Eq. 4). The original $X$ and $Y$ matrices are generated using the same parameters as in Fig. 6.

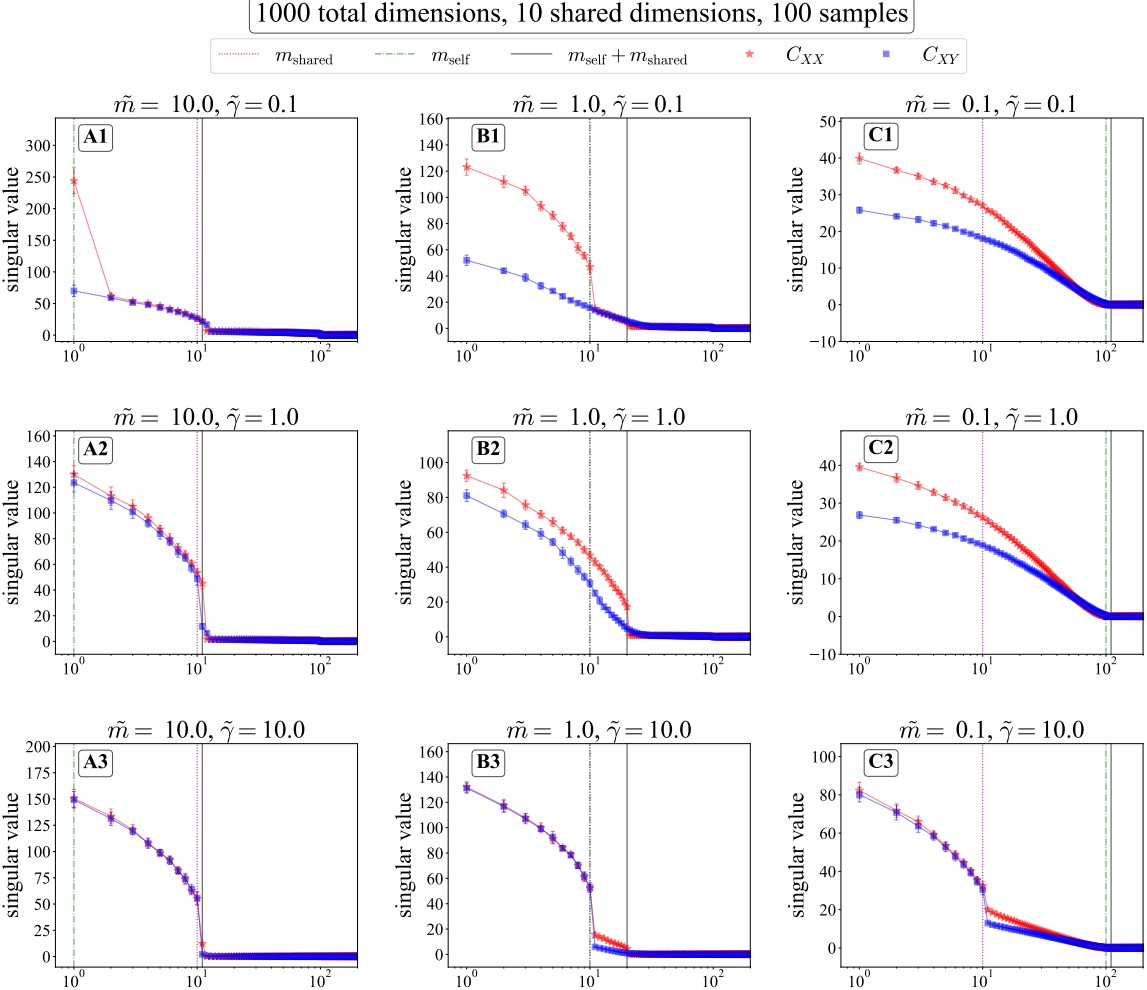

Figure 15: Singular values obtained from $C_{XX}$ and $C_{XY}$ (Eq. 4) for the parameters used in Fig. 6. Using two generated data matrices $X$ and $Y$ with $N_X = N_Y = 1000$, $m_{\text{shared}} = 10$ while we change $m_{\text{self}}$ to get the corresponding $\tilde{m}$, and $\gamma_{\text{self}} = 1$ while we change $\gamma_{\text{shared}}$ to get the corresponding $\tilde{\gamma}$. We calculate their corresponding $C_{XX}$ and $C_{XY}$ and plot their singular values versus their order for 100 samples. We observe similar behavior to Fig. 6, in the sense that if the shared signal is strong (Panels A3, B3, C3, where $\tilde{\gamma} = 10$), then we see 10 distinct singular values that correspond to the shared signal for all the different values of $\tilde{m}$. This is a regime where the results of applying IDR and SDR are equivalent (we even see the first 10 singular values are on top of each other). In Panel A1, we see one singular value of $C_{XX}$ standing out, followed by another 10 values of the shared signals, while $C_{XY}$ only identifies those 10 shared signals first. Panel A2 shows similar behavior, but the single self signal is now mixed among the others for $C_{XX}$. In Panel B2, while we see 10 distinct singular values of $C_{XX}$ appearing first, these correspond to the self signals, and the next 10 correspond to the shared signals, meaning that if we stopped in an IDR application after retaining 10 dimensions, we would not capture any shared signals. $C_{XY}$, on the other hand, shows a continuation of singular values without a gap, due to undersampling. Panel B2 shows similar behavior for $C_{XY}$, and for $C_{XX}$, we cannot even see a gap between the self and shared singular values. In Panels C1 and C2, the shared signal is overwhelmed by the self signals, and we cannot see any gap for $C_{XX}$ or $C_{XY}$ due to undersampling.

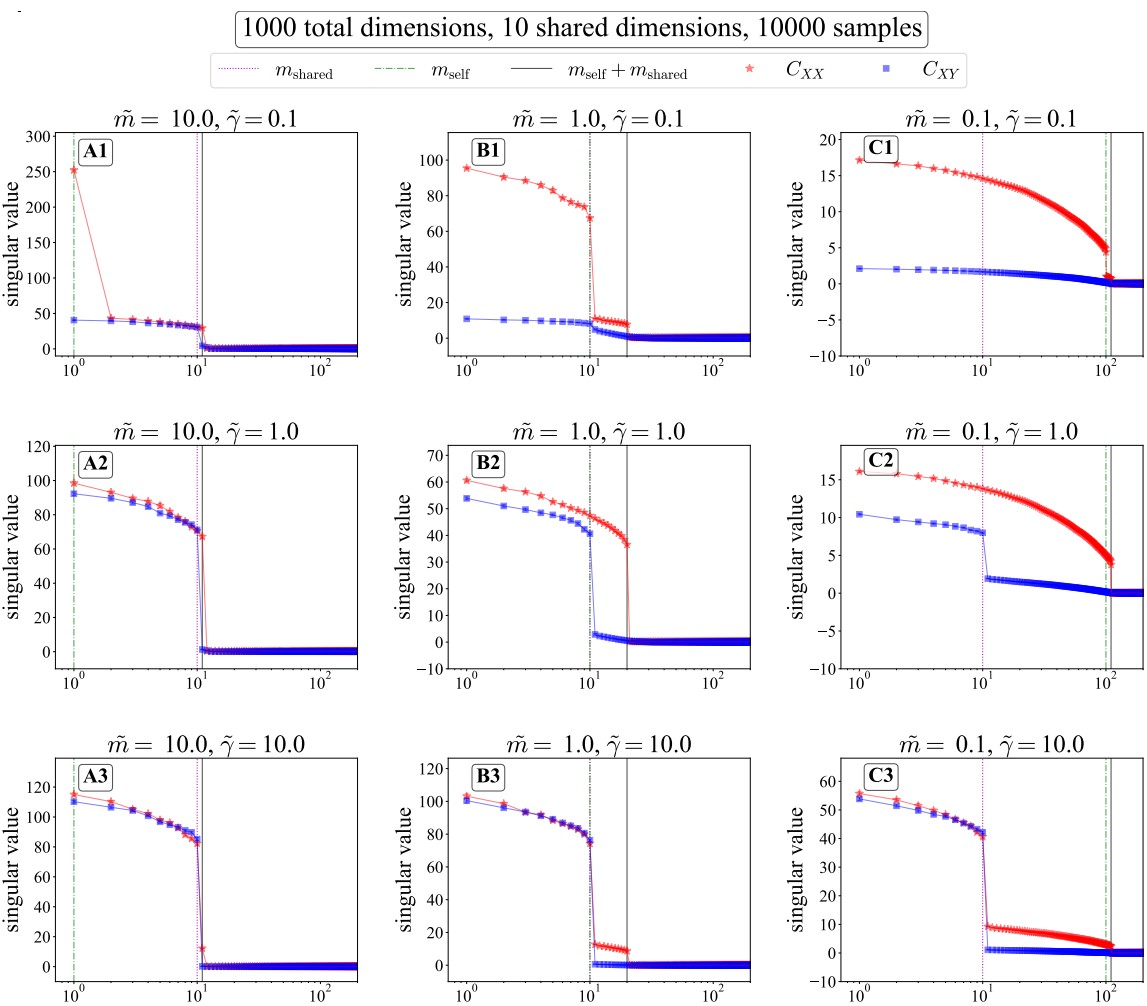

Figure 16: Similar to Fig. 15, but for 10000 samples – a well-sampled regime. We observe similar behavior to Fig. 15 in Panels A1, A2, A3, B3, and C3. In Panel B1, we now see a small gap in the spectrum of $C_{XY}$ that was not apparent in Fig. 15 due to undersampling. Panel B2 shows an even clearer gap for $C_{XY}$, whereas for $C_{XX}$, we still cannot see a gap between the self and shared singular values. Panel C1 shows similar behavior for $C_{XY}$, but for $C_{XX}$, we can see a gap after 100 singular values, followed by another small gap after an additional 10 singular values that correspond to the shared signal. However, if we are using an IDR method, this means we need to retain 110 dimensions, which is challenging to sample adequately. In Panel C2, we can now see a clear gap for $C_{XY}$ after 10 singular values corresponding to the shared signals, while we cannot see any gap for $C_{XX}$ because all the singular values for self and shared signals have equal power.

