# OpenReview forum: "Simultaneous Dimensionality Reduction: A Data Efficient Approach for Multimodal Representations Learning"
_TMLR — Accepted by TMLR_

### Review · Reviewer_CC1Y · 2024-02-21

**Summary Of Contributions:**

The authors present a numerical study of the relative performance of PCA compared to PLS, CCA and regularized CCA in extracting the shared variation from two noisy data views with partly shared and partly independent low-dimensional latent structure. Given the goal to denoise the cross-correlation matrix by dimensionality reduction, the results show that in the data model considered by the authors the shared dimensionality reduction methods PLS and (r)CCA outperform independent dimensionality reduction of the two views via PCA. The analysis shows that any independent variation of each view retained by PCA  leads to additional spurious correlations in the cross-correlation matrix for small sample size. This can be avoided by PLS and (r)CCA which directly select directions of shared variation. Of the methods compared, rCCA shows best overall performance. The authors also briefly discuss estimating the dimensionalities of shared and independent structure, and show an experiment with nonlinear data structure.

**Audience:**

Yes

**Broader Impact Concerns:**

/

**Claims And Evidence:**

Yes

**Requested Changes:**

## Main requested changes
1. **Clarification of model and task definition, scaling of low-rank terms**
The goal of the IDR/SDR analysis did not seem clearly operationalized (only implicitly through the definition of $\mathcal{RC}^\prime$, see point 2. below). The authors use both the terms "detection" and "reconstruction" of shared signals in the text. At least if the goal to study when the DR methods are able to detect the signal, the scaling of the low-rank signal terms should be explicitly discussed in the introduction or limitation sections: In the current strong scaling of the low-rank spikes, the signal should always be detectable in the proportional $T, N_{X/Y} \to \infty$ limit (except by CCA for $T/N < 1$). However, if scaling the variance of the low-rank entries with $N^{-1}$, a for each of the methods a BBP-type transition should occur, where detection becomes impossible. (I am aware of one reference where such a transition was studied for CCA, arXiv.2306.16393).
Concerning the model definition, eqs. (1,2), the surrounding text and footnote 2 on p.4,  slightly confusingly suggest that $X/Y \in \mathbb{R}^{N_{X/Y}}$ are single sample vectors, and correspondingly one would have  $R_{X/Y} \in \mathbb{R}^{N_{X/Y}}$, $U_{X/Y} \in \mathbb{R}^{m_{\textrm{self, }X/Y}}$, $P \in \mathbb{R}^{m_{\textrm{shared}}}$ vectors. However later the canonical version with $X/Y \in \mathbb{R}^{T \times N_{X/Y}}$ as data matrices is used.

2. **Motivation of $\mathcal{RC}^\prime$ and the design of the IDR-SDR comparison**
I was missing a discussion why $\mathcal{RC}^\prime$ captures precisely the goal of the data analysis. In particular, it seems to be only a fair metric if the goal is to extract the shared correlation removing any other variability, and it is very close to the quantity which CCA is optimizing. Would the performance difference between PLS and (r)CCA change if $\textrm{Cov}[Z_X, Z_Y]$ was used instead of $\textrm{Corr}[Z_X, Z_Y]$ ?  Furthermore, in the presence of strong (self- and shared-) signals, it is by design that PLS and (r)CCA outperform PCA on this metric. However, if $\geq m_{\textrm{self}} + m_{\textrm{shared}}$  dimensions are retained after denoising by PCA, the shared variability can still be extracted, for example by PLS, possibly with the same or better accuracy (or $Z_Y$ could be predicted from $Z_X$ using PLS-regression). In this sense it might be prudent to argue carefully in how far direct SDR is always preferable to using IDR as a pre-processing step.

3. **Regression step in the NIPALS algorithm**
In Appendix A, PLS is described as just performing SVD on $X^T Y$. However this is only true for the first singular value, afterwards the NIPALS algorithm deviates from the SVD result due to the regression steps on $X$ and $Y$ followed by deflation of $X^T Y$ (e.g. Wegelin, 2000, Tech Report 371, Univ. Washington).

## Additional changes

4. **Interpretation of performance results for PLS and rCCA**
It would strengthen the results of the paper to explain why PLS fails for small $T$ and $\gamma_{\textrm{shared}} > \gamma_{\textrm{shared}}$ (Fig. 3.B1), and why rCCA clearly outperforms both PLS and CCA in Fig.3 but also in Fig.8, even though as mentioned in Appendix A.1.4 it corresponds to interpolating between CCA and PLS when varying the regularization parameter.

5. **Test to determine the number of self- and shared signals in the data**
On p.11, varying $\tilde{q}$ by varying $|Z|$ is discussed as a possibility to determine the number of shared and independent signal components. This is an interesting suggestion. Does this method work better than obtaining $m_{\textrm{shared}}$ from thresholding the fraction of explained variance of $X^T Y$, e.g. from PLS or (r)CCA,  and $m_{\textrm{self}} + m_{\textrm{shared}}$ from thresholding the fraction of explained variance of $X, Y$ from PCA ?

6. **Typos / small changes**
   - abstract: "data set size" -> dataset size
   - In sect. 4.1 it is not mentioned that the SNR's of $X$ and $Y$ are symmetric (I assume?)
   - p.5: "we expect IDR to be more easily confused [...] by the self-signals". Could be made more precise
   - As a suggestion, it could be interesting to also show the singular value spectra of the underlying matrices for one or two relevant parameter choices

**Strengths And Weaknesses:**

## Strengths
- Numerical results on a data model which is simplified and interpretable, but practically relevant.
- Contribution to the development of best practices among practitioners of multi-view data analysis.

## Weaknesses
- Model and task definition are currently imprecise
- Unclear motivation of the performance metric $\mathcal{RC}^\prime$, and the design of the IDR-SDR comparison
- Limited mechanistic interpretation of the performance results for PLS and rCCA, and why rCCA performs best

---

> ### Author Response · Authors · 2024-08-07
> **Part 1**
>
> **We appreciate the thorough review and insightful comments regarding our manuscript. We are addressing the Reviewer's concerns and questions as follows:**
>
>
> ***Definition of the model***
>
> We thank the Reviewer for identifying possible confusion for how we defined the model in equations 1 and 2. We have clarified the main text and definition of the model in **Eqs.(1,2)** to make it explicit that the canonical versions of the data matrices $X/Y \in \mathbb{R}^{T \times N_{X/Y}}$ are used. We have clarified the dimensions and definitions of the other projection and data matrices accordingly.
> We have clarified the goal of SDR to detect shared signal.
>
> ***Scaling of low-rank terms***
>
> We observe a BBP-like transition in Figure 3A1 for PCA. Specifically along the y-axis at $\gamma_{\rm self}=0$, where there is only one signal (the shared signal). The strength of the self signal is modulated relative to the background noise $R_X$ (or $R_Y$), and is measured by $\gamma_{\rm shared}$. When $\gamma_{\text{shared}}\approx.35$ we see a transition from the signal being reconstructable to the signal being invisible behind the sampling noise. This is slightly different than the normal transition as here we observe it in the reduced matrices. In the original 1000 dimensional matrix, it is a normal BBP transition, where one singular value leaves the bulk. However, the transition is less pronounced in the reduced matrices due to their smaller size, and absence of self-averaging. We detect the transition with our  RC' metric, but we do not have the statistical power to explore its properties quantitatively. Since we cannot really explore the transition quantitatively, we chose not to focus on the transition specifically, but we now mention it in our discussion of **Fig.3**.
>
> ***Motivation of $\mathcal{RC'}$***
>
> We agree with the Reviewer that the choice of the $\mathcal{RC'}$ metric requires further elaboration. In response, we have included an additional discussion in **Section A.2**, which explores alternative options and provides more explanation for our choice. Specifically, we addressed the question of using covariance versus correlation. We chose correlation due to its utility and intuitiveness, as it provides a consistent metric across all scenarios. If we were to adopt a covariance-based metric, we might achieve slightly better accuracy for PLS, as discussed in more detail in the text. However, we would like to emphasize that our main objective is to demonstrate the effectiveness of SDR compared to IDR, which is evident when comparing PLS to PCA. While the comparison between PLS and CCA is interesting, it is not the primary focus of our study.
>
> ***The design of IDR-SDR comparison***
>
> We fully agree with the Reviewer that there are situations where SDR may not perform better than IDR. For instance, as mentioned, this may be true when keeping $\geq m_{\text{self}} + m_{\text{shared}}$ dimensions, which we highlighted in several instances, including **Fig.6**. However, adding more dimensions than necessary to account for $m_{\text{self}}$ incurs an additional cost in terms of the number of samples needed to estimate those extra dimensions. This issue becomes more critical if $m_{\text{self}} \gg 1$, as demonstrated in **Fig.6C2**. Therefore, while we acknowledge that SDR is not universally superior, it does offer advantages in certain scenarios when searching for shared representations, as discussed in the text. Importantly, in our tests, SDR has never been worse than IDR, which we highlight.
>
> ***Regression in NIPALS algorithm***
>
> We appreciate the Reviewer's attention to the distinctions between PLS computed via NIPALS and SVD. We have further clarified these differences in **Section A.1.2**. Specifically, the PLS implementation described in the appendix, which performs deflation after calculating each component, is often referred to as *PLSCanonical*. Additionally, an alternative simpler approach based on direct SVD calculation of the covariance matrix is often known as *PLSSVD* (As in *Scikit-learn: Machine Learning in Python}, Pedregosa. et al., 2011*, for example).

---

> ### Author Response · Authors · 2024-08-07
> **Part 2**
>
> ***Interpreting performance of PLS and rCCA***
>
> When $T < N_X$, CCA fails because the matrices $C_{XX}$ and $C_{YY}$ are rank-deficient and thus not invertible, as we emphasize in the text. In this scenario, rCCA enables proper inversion of these matrices, allowing (the regularized version of) CCA to function. Conceptually, the main difference between PLS and CCA is that PLS does not enforce orthogonality among the weights $w_X^{(i)}$ and $w_Y^{(i)}$ that diagonalize $C_{XX}$ and $C_{YY}$, whereas CCA does. For instance, while two pairs of singular vectors $(w_X^{(1)}, w_Y^{(1)})$ and $(w_X^{(2)}, w_Y^{(2)})$ are mutually orthogonal as pairs, $w_X^{(1)}$ and $w_X^{(2)}$ are not orthogonal to each other. This partial overlap can result in signals not being fully expressed in these directions, reducing the total correlation compared to CCA, where each direction $w_X^{(i)}$ and $w_Y^{(i)}$ is orthogonal to every other direction, allowing each signal to fully occupy these directions and maximizing total correlation. Therefore, our goal is to make CCA work, but due to inversion issues, rCCA serves as a substitute. This is evident in Figures 1-5, where, under good sampling conditions, rCCA and CCA yield almost identical results. This whole discussion is now a part of the **Results** section of the manuscript.
>
> ***On determining the number of self and shared signals***
>
> We thank the reviewer for their interesting idea. We think that thresholding the fraction of explained variance in $X^T X$ vs $X^T Y$ is conceptually (and probably practically) similar. However, we believe that calculating the correlation between $Z_X$ and $Z_Y$ and observing its peak might be more informative in some situations compared to simply observing the explained variance. The explained variance might be long-tailed (i.e., have many components that contribute minimally but not small enough to create a clear gap for thresholding), and thus correlation might provide better insight. Additionally, with thresholding, we do not impose a cut off on the noise, like when we do with subtracting $\mathcal{RC_0}$, which might lead to additional ambiguity in estimating the number of shared and independent signals, especially in undersampled situations. In view of this discussion, we chose not to make any changes to the text.
>
> ***Additional figures for singular value spectra***
> We now provide two additional supplementary figures (**Figs.15-16**) in **Section A.6**, where we show the singular value spectra of the data matrices $X^T X$ and $X^T Y$ for various sampling conditions (under-sampled and well-sampled), and different number of self signals and different strengths of the shared signals (similar to Fig.6).
>
> ***Difference between reconstruction and detection***
>
> We agree with the Reviewer that we have been somewhat sloppy in our use of terminology. In general, the terms "reconstruction'' and "detection'' mean different things. However, in the context of our model, the signal is indeed the matrix product $PQ$, and its detection via SVD-based method is equivalent to its reconstruction, as both mean that we obtain a correlation between $Z$s and the signal itself. We now comment on this in the text.
>
> ***Changes to text for small typos / small changes:***
>
> We thank the Reviewer for pointing out typos and minor corrections. We have incorporated them all in our revised manuscript.

---

> ### Comment · Reviewer_CC1Y · 2024-08-09
> **Thanks for the revision**
>
> I would like to thank the authors for their reply and work on the revision, especially also for adding the supplementary plots of the singular value spectra. Many of my comments have been addressed.
>
> Concerning the existence of a BBP transition, I'd first like to stress that as expected from the strong scaling of the signal, in the case of Fig1 (A1 and A2) visibly no BBP transition takes place as for $\gamma_\mathrm{self} = 0 $ the shared direction can be found all the way to $\gamma_\mathrm{shared} \ll 1 $.
> Now, I understand from the authors' response and the plot that it could be argued that in the setting of Fig3, with a large number of self-signals and one shared signal, the self-signals form a second bulk of singular values (different from the noise bulk), and that a singular value corresponding to the lone shared signal may be observed to separate from this bulk of the self signals.  I am not immediately sure whether the observed transition line would be stationary or still move when $N,T \to \infty$. One might expect that this transition becomes a sharp phase transition when the also the number of self signals forming the bulk is sent to infinity with $N,T$ (in which case the relative strength of the original noise would vanish). I think that clarifying and discussing these scaling limits would strongly benefit the reception of the paper by more theoretical researchers.
>
> Related to Fig3 A1 and A2, along the y-axis where  $\gamma_\mathrm{self} = 0 $, there should be no self signals in the data, with the parameters otherwise as in Fig1; why are the results along this line in Fig3 different than in Fig1 A1 and A2?

---

> > ### Author Response · Authors · 2024-08-12
> >
> > We sincerely thank the reviewer for their prompt response and their constructive comments on our work.
> >
> > Regarding the final point about the discrepancy between Figures 3A and 1A, the values in question start with $\gamma = 0.05 \rightarrow 0$, resulting in a minor contribution from the self (and shared on the other axis) signals. This contribution is negligible in Figure 1 (due to the presence of just one additional signal) but leads to a noticeable difference in Figure 3 (due to the cumulative effect of one signal multiplied by 30). We have added to the clarifications footnote in the text to address this point.
> >
> > For the BBP transition discussion, it is important to note that we are not in the classical limit where $T, N \rightarrow \infty$. Additionally, we are not directly observing the spectrum of the singular values of the matrices. Instead, we observe the correlation between the embeddings, which corresponds to the singular value when only one dimension is retained. This correlation is further corrected by baseline noise (which we can think of as subtracting the effect of the bulk of the noise). Once more than one dimension is retained, the direct mapping between the observed correlations and the distribution of the singular values becomes unclear.
> >
> > We agree that the analysis of BBP transitions in this setup is an intriguing limit worth considering, but is beyond the scope of our current work.

---

> > > ### Comment · Reviewer_CC1Y · 2024-08-13
> > >
> > > Thank you very much for the response.
> > >
> > > While I do think that it would be useful to state the high-dimensional limit which the authors are envisioning with their model, that the transitions in Figs 3 and 4 could be largely understood based on the singular value spectra of the covariance matrix (for PLS) and the normalized correlation matrix (for CCA), and that such a discussion would strengthen quite significantly the experimental results and conclusions by providing a mechanistic understanding for this basic part of the results on the data model (and thereby its wider generality and  possible limitations), I of course respect their choice not to pursue this direction.
> > >
> > > From my perspective, the main technical concerns of correctness and clarity have been addressed and I would, subject to discussion with the other referees, be happy to recommend the current revision for acceptance.

---

### Review · Reviewer_xNmU · 2024-06-26

**Summary Of Contributions:**

The authors study several methods for linear dimensionality reduction on multi-modal data sets.
- Principal component analysis (PCA) as an example for Independent Dimensionality Reduction (IDR), where each data modality is compressed independently;
- Partial Least Squares (PLS) and Canonical Correlations Analysis (CCA) as examples of d Simultaneous Dimensionality Reduction (SDR), where one simultaneously compresses the modalities to maximise the covariation between the reduced descriptions, while paying less attention to how much individual variation is preserved.
Their goal is in particular to contrast independent with simultaneous dimensionality reduction techinques.

To carefully analyses the properties of the various methods, the authors study their performance on a synthetic model of multi-modal data. Their idea is to generate two data matrices, $\tilde X$ and $\tilde Y$, which have three parts: an independent white noise; a lower-rank signal that is specific to each matrix, $U_X V_X$ and likewise for $Y$; and finally a shared lower-rank signal that is partly shared across the two data matrices: $P Q_X$ vs. $P Q_Y$. They evaluate the methods by measuring how well the singular directions recovered by each method recovers the shared signals in the data. They broadly find that methods that do a simultaneous decomposition of two data sets perform better at recovering the shared structure than methods that decompose the two datasets independently. Qualitatively similar results are obtained on a non-linear example (noisy MNIST)

**Audience:**

Yes

**Broader Impact Concerns:**

None.

**Claims And Evidence:**

Yes

**Requested Changes:**

I think the paper is publishable as is.

**Strengths And Weaknesses:**

## Strengths

- The paper performs a careful evaluation of several linear dimensionality reduction methods. Given their prevalence in machine learning, for example in the analysis of representations of trained neural networks, or in the analysis of large-scale neural recordings, such a careful evaluation is welcome.
- The paper carefully studies several regimes of interest, including over- and undersampled regimes.
- The paper is clearly written and well-structured.

## Weaknesses

As the authors acknowledge themselves, the restriction to a linear model of data is the main limitation of the present study. However, I believe that this study is an excellent starting point for an analysis of the algorithms on non-linear models of data, of which I would like to add models of the ICA flavour to the ones mentioned by the authors in their conclusions!). Furthermore, analysing the performance of various algorithms in extracting low-rank signals from data is a very active research area, and the present model might inspire a more mathematical analysis of the performance.

I therefore believe that this study, while being undoubtedly technically sound, should attract the interest of  some individuals in TMLR's audience, and therefore be published in TMLR.

---

> ### Author Response · Authors · 2024-08-07
>
> We thank the Reviewer for their support, review, and insightful comments. We have expanded our **Discussion** to address other methods (like ICA, NMF, etc.) that fall within the IDR and SDR classes, and pointed out the need to have generative models, to study when such approaches are expected to work well. We expect ICA and NMF to behave similarly to the IDR methods we analyzed in the paper - they will be able to detect shared signals when the latter are stronger than the self signal (though the measure of strength will not necessarily be a simple variance). We further expect methods such as Cross-modal Factor Analysis and Deep Variational Symmetric Information Bottleneck to behave similar to the SDR methods we examined by identifying the shared signal first (again, with the performance metric being different).

---

> > ### Comment · Reviewer_xNmU · 2024-08-18
> >
> > Thank you for your additional explanation. I have now read the other reviews and the rebuttals, and I agree that the revised paper can be published in its current form.

---

### Review · Reviewer_ngHm · 2024-08-01

**Summary Of Contributions:**

The paper conducts a series of numerical experiments comparing independent dimensionality reduction (IDR) and simultaneous dimensionality reduction (SDR) methods for analyzing high-dimensional, multimodal datasets.

In the first set of experiment, using a linear generative model, the authors demonstrate that SDR methods, like Partial Least Squares (PLS) and Canonical Correlations Analysis (CCA), outperform IDR methods such as Principal Components Analysis (PCA) in reconstructing covariation structures, especially in small sample regimes. Moreover, the authors investigated the performance of regularized CCA (rCCA), for which the definition involves a convex interpolation problem between PLS and CCA, and it was showed that rCCA excels particularly in undersampled scenarios by maintaining nearly perfect reconstruction of shared signals even when self signals and noise are present.

The second set of experiments explores the performance of linear dimensionality reduction (DR) methods on nonlinear data using a noisy MNIST-inspired dataset. The dataset consists of digit images with variations such as random rotations, scaling, and background noise. The study compares methods like PCA, PLS, and rCCA, focusing on their ability to detect shared signals amidst view-specific noise, which is evaluated by calculating the total corrected correlation between the obtained low-dimensional representations and the inverse of the number of samples per retained dimensions.
Results show that, in the undersampled scenario with only 1000 samples, rCCA and PLS show an earlier detection of shared signals (measured by the number of retained dimensions after reduction) compared to PCA, which initially falls behind. However, as the number of dimensions grows, all methods experience a reduction in correlation, attributed to the higher noise levels caused by having fewer samples per dimension. While, in the oversampled scenario with 10,000 samples, all methods initially show increased correlation with more retained dimensions, then decline as more singular vectors are estimated from the same sample size, with rCCA consistently outperforming the other methods.

**Audience:**

Yes

**Broader Impact Concerns:**

-

**Claims And Evidence:**

Yes

**Requested Changes:**

- A summary of key takeaways would facilitate reading the paper.
- Some theoretical insights or geometric intuitions would be helpful in making the paper easier to comprehend.

**Strengths And Weaknesses:**

Overall, the article is well-written, although a bit verbose. Furthermore, it addresses a significant question to the community. In particular, the article underscores a notable phenomenon in rCCA performance: the performance curve exhibits a peak as a function of the inverse of the number of samples per retained dimension. However, the lack of theoretical insights or geometrical intuitions made the content difficult to follow and hindered the ability to draw definitive conclusions.

---

> ### Author Response · Authors · 2024-08-07
>
> We thank the Reviewer for their suggestions and recognition of the importance of evaluating linear DR methods for their scaling properties with samples and properties of the underlying dataset being examined (number and strength shared signals compared to self signals). While we recognize that our paper is verbose in places, we wanted it to be accessible to a broader audience. We have added a bullet point list in the **Introduction** where we have identified our major contributions and an additional section to the **Discussion** summarizing the major results. We have also added a back-of-the-envelope scaling argument to the **Results** to give an intuition for the optimal number of reduced dimensions.

---

> > ### Comment · Reviewer_ngHm · 2024-08-13
> >
> > I would like to thank the authors for their answer. Is it possible to highlight the changes in a different color (during the period of discussion)?

---

> > > ### Author Response · Authors · 2024-08-13
> > >
> > > We thank the Reviewer for their prompt response.
> > >
> > > To view the differences between the various document versions, OpenReview provides a built-in PDF comparison tool that you can access by following these steps:
> > >
> > > 1. On the original submission page, under the title at the top, you will see a section labeled **"Revisions"**.
> > >
> > > 2. Click on **"Compare Revisions"**.
> > >
> > > 3. From there, you can choose which versions of the document to compare. Select the first submitted version at the bottom of the list and the most recent version (which includes all the incorporated reviews).
> > >
> > > 4. After selecting the versions, click **"Review Difference"**.
> > >
> > > 5. As you scroll down, you will see two generated views highlighting the differences between the selected versions.
> > >
> > > 6. You can toggle between different views under the **"View"** tab to see the edits either in the most updated version, or both versions side by side.
> > >
> > > For your convenience, here is a direct link to the document comparison: (https://api.draftable.com/v1/comparisons/viewer/yVmSPr/gdDpTgXjyIES?valid_until=1723565658&signature=194da27af7f65ad9c89575b197fb20b472cfba3c5ae5bc227ed2ff9a5156cea2&wait).
> > >
> > > Please do not hesitate to reach out if further clarification is needed.

---

> > > > ### Comment · Reviewer_ngHm · 2024-09-06
> > > >
> > > > Thank you. I believe the revised paper is ready for publication in its current form.

---

### Author Response · Authors · 2024-08-07

We thank the Reviewers for their time and effort and their general positive sentiment towards our work. We found their comments very useful in enhancing our paper, and we considered and incorporated all of them. The paper has been updated accordingly, and we have also responded to each reviewer's comments individually.

---

### Decision · Action_Editor_3SYK · 2024-09-09

**Recommendation:** Accept as is

**Comment:**

The decision to accept the paper is based on its overall technical soundness, clear writing, and valuable contributions to understanding linear dimensionality reduction methods. While the reviewers note some areas for potential improvement, such as providing more theoretical insights and considering non-linear models, they generally find the paper well-prepared and relevant.

**Audience:**

The reviewers believe that the paper is relevant and would be of interest to the TMLR audience. The topic of linear dimensionality reduction methods and their performance in various data regimes is significant for machine learning practitioners and researchers, particularly those working on multi-modal data analysis or interested in dimensionality reduction techniques.

**Claims And Evidence:**

The reviewers agree that the claims made in the paper are supported by accurate, convincing, and clear evidence. They acknowledge the thoroughness of the experiments conducted and the soundness of the results presented. However, there are suggestions for further theoretical insights or explanations to enhance the understanding and support of some of the empirical findings.